



# Open system pingos as hotspots for sub-permafrost methane emission in Svalbard

Andrew J. Hodson[1,2], Aga Nowak[1], Kim Senger[1], Kelly Redeker [3], Hanne H. Christiansen[1], Søren Jessen[4], Mikkel T. Hornum[1,4], Peter Betlem[1], Steve F. Thornton[5], Alexandra V. Turchyn[6], Snorre Olaussen[1], Alina Marca[7]

1Department of Arctic Geology, University Centre in Svalbard (UNIS), N-9171 Longyearbyen, Norway.
[2]Department of Environmental Science, Western Norway University of Applied Sciences, Røyrgata 6, N-6856 Sogndal, Norway
[3]Department of Biology, University of York, YO10 5DD, UK.
[4]Department of Geosciences and Natural Resource Management, University of Copenhagen, 1350 Copenhagen K, Denmark.
[5]Department of Civil and Structural Engineering, University of Sheffield, S10 2TN, UK.
[6]Department of Earth Sciences, University of Cambridge, CB2 3EQ, UK.
[7]School of Environmental Sciences, University of East Anglia, Norwich, NR4 7TJ, UK.

*Correspondence to*: Andrew J. Hodson (AndrewH@unis.no)

## Abstract

Methane release from beneath lowland permafrost represents an important uncertainty in the Arctic greenhouse gas budget. Our current knowledge is arguably best-developed in settings where permafrost is being inundated by rising sea level, which means much of the methane is oxidised in the water column before it reaches the atmosphere. Here we provide a different process perspective that is appropriate for Arctic fjord valleys, where local deglaciation causes isostatic uplift to out-pace rising sea level. We show how the uplift induces permafrost aggradation in former marine sediments, whose pressurisation results in methane escape directly to the atmosphere via ground water springs. In Adventdalen, Central Spitsbergen, we show how the springs are historic features, responsible for the formation of open system pingos, and capable of discharging brackish waters enriched with high concentrations of mostly biogenic methane (average 18 mg $L^{-1}$). Thermodynamic calculations show that the methane concentrations sometimes marginally exceed the solubility limit for methane in water at 0 °C (41 mg $L^{-1}$). In our case study, emissions from just four pingo springs with a combined discharge of less than 2 L $s^{-1}$ increase the land-atmosphere methane flux by 16 %. This confirms that sub-permafrost methane migration deserves more attention for improved forecasting of Arctic greenhouse gas emissions.

## 1    Introduction

Methane evasion to the atmosphere from thawing Arctic permafrost represents a significant risk to future greenhouse gas management, and so great emphasis has been placed upon quantifying the release of methane from the active layer during summer thaw (Schuur et al., 2015). However, the potential for methane evasion from deeper sub-permafrost sources also





exists (Anthony et al., 2012; Betlem et al., 2019; Kohnert et al., 2017), but since the means by which the gas by-passes the permafrost are unclear, their possible timing, magnitude and impact are very uncertain. Recent research has provided significant insights into the role of landscape change and methane release from low relief Arctic shelf environments typical

of the Canadian, Siberian and North Alaskan coastlines (Kohnert et al., 2017; Frederick et al., 2016; Dmitrenko et al., 2016). Here, sea level inundation has enhanced methane escape by inducing permafrost thaw (Frederick et al., 2016). However, this mechanism is not relevant to many fjord coastlines in the Arctic because isostatic uplift has out-paced sea level rise (Dutton et al., 2015). Here, the uplift of sediments deposited in the fjord since the Last Glacial Maximum (LGM) has caused their exposure to the atmosphere, resulting in a period of freezing and permafrost aggradation (e.g. Cable et al., 2018; Gilbert et

al., 2017, Gilbert et al., 2018). Fjord coastlines which have undergone significant isostatic uplift are typical of Svalbard, Novaya Zemlya, northern Greenland and the Canadian Arctic archipelago. It is therefore significant that these areas are poorly represented in our current understanding of pan-Arctic methane emissions from the land surface.

Fjords are notable for some of Earth's most rapid rates of sedimentation and organic carbon burial during glacial retreat (Smith et al., 2015; Syvitski et al., 1986; Włodarska- Kowalczuk et al., 2019), producing thick sediment sequences

conducive to biogenic methane production. In addition, the rocks underlying many Arctic fjords support either proven or highly probable natural gas resources. Therefore methane from geogenic sources such as coal beds and shale is also likely to be present. At the LGM, widespread methane hydrate stability zones were present under the ice sheets, providing a transient reservoir for both the biogenic and geogenic methane. The warmer period that caused the onset of ice sheet retreat after the LGM caused the gas hydrates to become thermodynamically unstable, and the methane began to escape rapidly through the

recently uncovered sea floor (Crémière et al., 2016; Smith et al., 2001; Weitemeyer and Buffet, 2006). Evidence for such rapid fluid escape include pockmarks (Crémière et al., 2016; Portnov et al., 2016) (Fig. 1a), whose occurrence in Svalbard is particularly well-documented because some of them remain active today (Liira et al., 2019; Sahling et al., 2014). Sea floor methane emissions are subject to very significant removal processes due to dissolution and oxidation within the overlying water column (Mau et al., 2017; Sahling et al., 2014).  Further, Pohlman et al. (2017) have shown that sea floor gas

emissions in coastal waters off Svalbard may also be offset by far greater rates of atmospheric $CO_2$ sequestration into the overlying surface waters, because the rising bubbles help nutrient-rich bottom waters rise up to fuel the photosynthesising plankton community. However, Hodson et al. (2018) showed that pockmarks exposed by isostatic uplift have the potential to form methane seepage pathways on land. Since any groundwater carrying the gas through the permafrost will be subject to freezing temperatures, these features are likely to become discernible as small, ice-cored hill forms known as open system

pingos (Fig. 1b). Pingos and other terrestrial seepages therefore must be considered as migration pathways through what is otherwise regarded as an effective seal or "cryospheric cap" formed by the permafrost (Anthony et al., 2012). Such routes arguably represent the most harmful greenhouse gas emission pathway for methane trapped beneath permafrost, because gas can escape directly to the atmosphere without removal by oxidation within an overlying water column.

This paper therefore investigates how methane readily escapes from beneath permafrost by exploiting the open

system pingos that have formed following isostatic uplift and permafrost aggradation in Svalbard fjords. We show that the

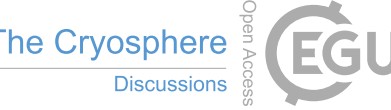

pingos form natural "hotpots" for the ventilation of sub-permafrost methane directly to the atmosphere, accounting for a meaningful proportion of the total annual methane emissions in Adventdalen, a representative, well-researched fjord valley system in Central Spitsbergen, Svalbard.

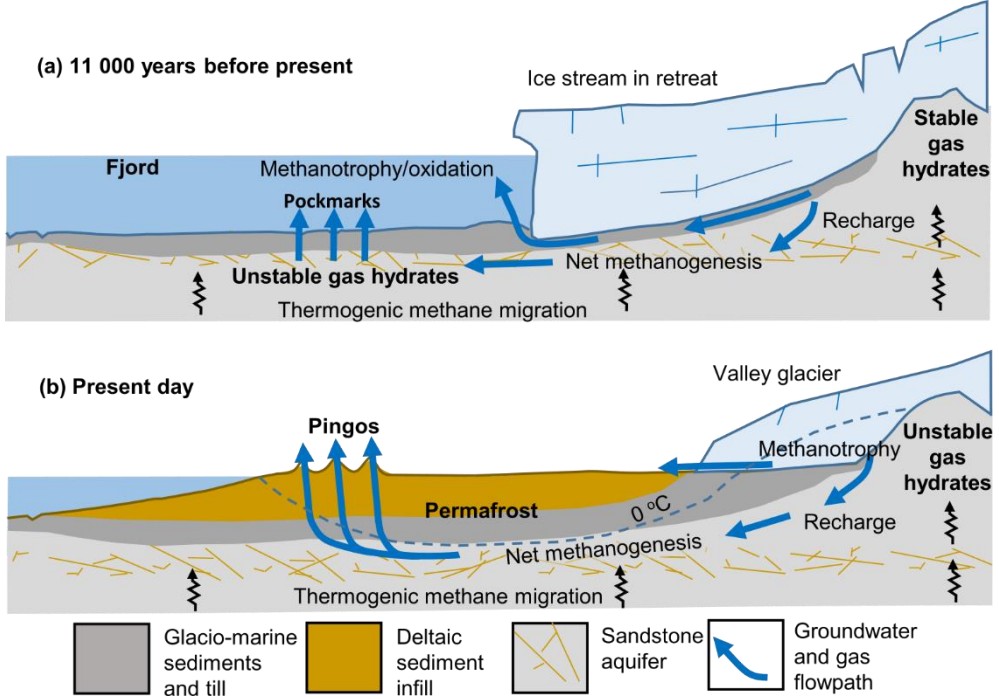

**Figure 1.** Landscape change and methane migration pathways. **(a)** During deglaciation after the Last Glacial Maximum ca. 11 000 years ago. **(b)** Today, following isostatic uplift and permafrost aggradation in a deltaic sediment infill overlying glacial till. The conceptual model of landscape change was based upon Gilbert et al. (2018).

## 2 Methods

### 2.1 The field site

Adventdalen's open system pingos are located in a lowland valley that has been rapidly in-filled by a pro-grading delta system throughout the Holocene. This was driven by ice sheet retreat commencing ca. 11 000 years ago (Gilbert et al., 2018) and is represented by the landscape model in Fig. 1. As with many open system pingos in Central Spitsbergen, their formation was intricately linked to changes in groundwater dynamics that occur after such deltaic sediments emerge from below sea level and start to freeze. This permafrost aggradation increases hydraulic pressure and thus forces residual groundwaters toward the land surface. Further freezing results in expansion and the formation of a small hill with an ice core, or "pingo" up to 40 m higher than the surrounding topography (Liestøl, 1996; Yoshikawa et al., 1995; Yoshikawa,



1993). Since the hydraulic conductivity of the fine-grained, uplifted marine sediments is very low (Hornum et al., In Review), the fluids are likely to exploit any former pockmarks that are uplifted with them.

Four of six open system pingos in Adventdalen discharge groundwaters all year (Fig. 2). These springs were sampled before the melt season using drilling techniques that allowed direct access to the emerging waters through a winter ice lid up to 2 m thick. During summer, surface meltwater flooding in the valley hinders access to the pingos. Figure 2 shows that two pingos (Lagoon Pingo and Førstehytte Pingo) are situated in the lower part of the valley, whilst two others (Innerhytte Pingo and River Bed Pingo) are up-valley, adjacent to the former marine limit at ca. 70 m above sea leve. Like many fjord valleys, the rate of sedimentation was extremely high during ice sheet retreat, and so up to 60 m of valley in-

filling has occurred (Cable et al., 2018; Gilbert et al., 2018). However, the permafrost in the valley floor of Adventdalen is up to 120 m thick, so much of the fine sediments have frozen since their exposure by isostatic uplift during the Holocene, with the exception of the sediments closest to the contemporary shoreline and pockets of saline "cryopegs" further up-valley (Keating et al., 2019). Lagoon Pingo, nearest to the coast, is thought to be less than 200 years old, and has springs (presently three) that have been documented as early as 1926 (Liestøl, 1996; Yoshikawa and Nakamura, 1996). At Førstehytte Pingo, a

spring has also been known to exist since the 1920's, but the pingo is thought to be much older. Radio-carbon dates for molluscs in the marine sediments uplifted by the Førstehytte Pingo give a maximum age limit of 7000 $\pm$ 70 years (Yoshikawa, 1993; Yoshikawa and Nakamura, 1996). Innerhytte Pingo and River Bed Pingo are of unknown age, and since they lack a cover of marine sediments containing mollusc shells, no radiocarbon dates are available.

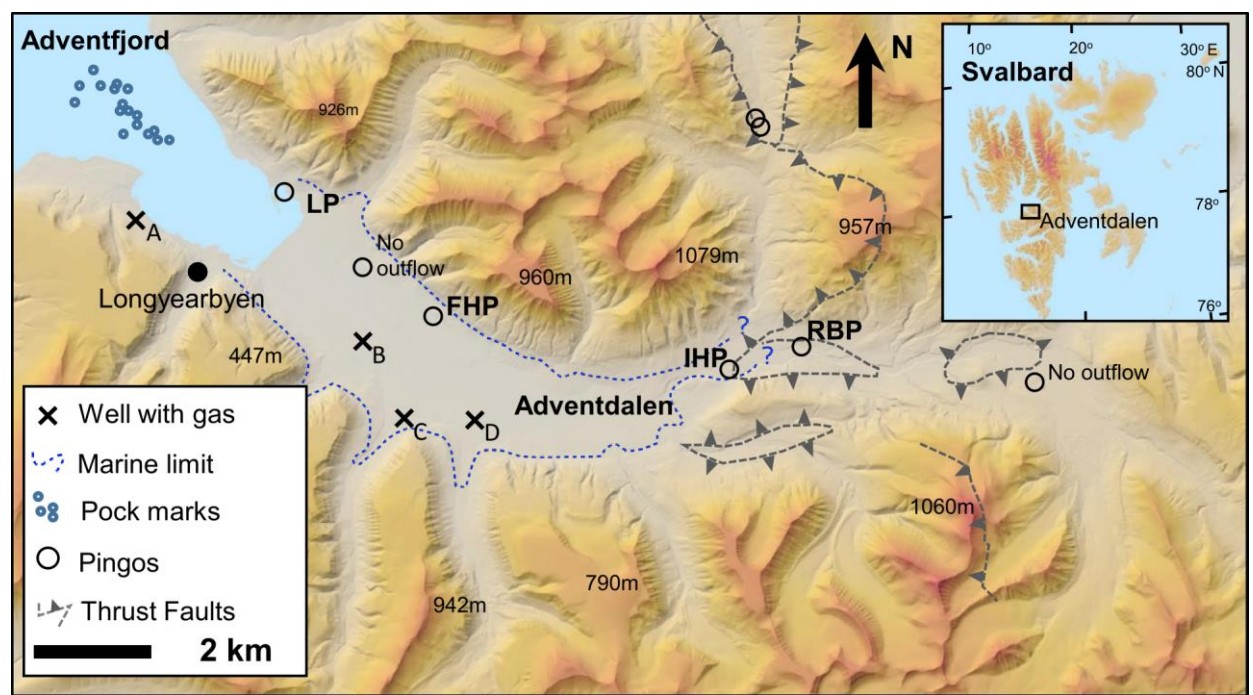



**Figure 2.** Adventdalen and its open system pingos. Active springs exist at Lagoon Pingo, Førstehytte Pingo, Innerhytte Pingo and River Bed Pingo (LP, FHP, IHP and RBP respectively). Wells C and D are part of the UNIS $CO_2$ Well Park (Braathen et al., 2012). Map developed online at www.svalbardkartet.npolar.no.

## 2.2 Fieldwork

Field work involved consecutive spring time sampling campaigns (March - April) at the four sites from 2015 until 2017. In addition, opportunistic sampling was conducted in summer 2017, where low river levels made access to the field sites possible. We focussed our sampling on the larger, discrete springs that were clearly associated with the pingo on account of their proximity. However, more diffuse springs are continuously being discovered in the vicinity of the pingos whose influence on methane emission to the atmosphere are considered later in the paper. Future work will integrate the point source and diffuse seepages to provide a landscape perspective.

Pingo springs were sampled after drilling up to three metres through icings that had formed by continuous discharge from the pingos during winter. This involved use of a 7 cm diameter Kovacs drill and Stihl two-stroke engine. Although the icing surfaces were sometimes visibly cracked, with outflow of water, drilling was still employed to reduce the likelihood of oxygenation before sampling and contamination from local snow.

At the sampling site, pH, temperature, dissolved oxygen and Oxidation-Reduction Potential (ORP) were recorded using Hach Lange HQ 40D meters and dedicated electrodes/sensors. These were calibrated prior to use with the exception of the dissolved $O_2$ measurement, which was conducted using the luminescence method and thus used a factory calibrated sensor tip. To prevent freezing problems and electrode malfunction, water samples were pumped through a bespoke, air-tight flow cell with an internal heating element maintaining the sample flow at ca. 7 $^{\circ}$C.

## 2.3 Analytical work

Samples for dissolved iron and manganese analysis were syringe-filtered immediately in the field through 0.45 μm filters into pre-cleaned 15 mL Eppendorf Tubes, before acidification to pH ~ 1.7 using reagent grade $HNO_3^-$ (AnalaR 65 % Normapur, VWR, IL, USA). The analysis of dissolved Fe and Mn was then completed using Inductively Coupled Plasma Mass Spectrometry or ICP-MS (PerkinElmer Elan DRC II, MA, USA). Precision errors of the analyses were < 5 % according to repeat analyses of mid-range standards, with a detection limit of 1.0 μg $L^{-1}$. No contaminants were detected above this limit in the analyses of blank deionised water samples. Samples for major ion analysis (here $Ca^{2+}$, $Mg^{2+}$, $Na^+$, $K^+$, $Cl^-$, $NO_3^-$, $SO_4^{2-}$) were also filtered in the same manner (but not acidified) and stored in 50 mL Corning centrifuge tubes after being triple rinsed with filtrate. The analysis was conducted on Dionex DX90 Ion Chromatographs with a detection limit of 0.02 mg $L^{-1}$ for the lowest, undiluted analysis. Precision errors for these ions were all <5 % for mid-range standards.

Charge balance calculations were used to provide the indicative values of $HCO_3^-$ and $CO_3^{2-}$, given (as DIC or dissolved inorganic carbon) in Table 1. Excess $CO_2$ levels were estimated from calculations of the partial pressure of $CO_2$ using the online WEB-PHREEQ Geochemical Speciation Software (https://www.ndsu.edu/webphreeq/).



Samples for the determination of dissolved methane and carbon dioxide concentrations as well as $\delta^{13}$C-CH$_4$ and $\delta^{13}$C-DIC of the waters were taken directly from the spring following immersion, complete filling and sealing of a 22 mL Wheaton bottle with a crimp-top lid with septum. The samples were stored inverted under water at 4°C until analysis. The analysis of the CH$_4$ was performed by gas-chromatography on a Shimadzu GC-2014 instrument equipped with a methaniser and flame ionisation detector, using a 30 m GS-Q, 0.53 mm internal diameter column with N$_2$ as a carrier gas at a flow rate of 8 mL min$^{-1}$, and injection, oven and detector temperature of 60, 40 and 240 °C, respectively. The sample size was 100 μL and the sample run time was 3 minutes at 40 °C. Concentrations of dissolved CH$_4$ were obtained according to a mass balance calculation for the samples (McAuliffe, 1971), in which a known volume of N$_2$ was injected into sample vials to create a headspace (Tyler et al., 1997). After shaking and equilibration (2 h) the CH$_4$ partitioned into the headspace was analysed by GC-FID and the corresponding mass in the gas and aqueous phase was determined by Henry's law, to obtain a final concentration in the water sample. Calibration gas standards were prepared by serial dilution of certificated 60 % CH$_4$: 40 % CO$_2$ mixed gas using O$_2$-free N$_2$ as the balance gas. The method detection limit was 10 ppm for CH$_4$ in the headspace sample and the analytical precision error was < 5 %.

Analysis of dissolved methane isotopic composition and concentration was performed using the gas headspace equilibration technique (Magen et al., 2014) (5 mls sampled water were injected into a Viton-stoppered, He-flushed 120 mL glass serum vial). 10 mls of the headspace was then flushed through a 2 mL sample loop, and injected onto a 25 m MolSieve column within an Agilent 7890B GC attached to an Isoprime100 Isotope Ratio Mass Spectrometer (IRMS) (Tyler et al., 1997). Analytical precision errors for samples > 3 ng-C were better than 0.3‰ for isotopic values, and < 3.5 % for concentration, based on methane standard injections. $\delta^{13}$C$_{DIC}$ was measured by a Continuous Flow Isotopic Ratio Mass Spectrometer (Thermo-Finnegan Delta V with gasbench interface) and an error of 0.1‰. All $\delta^{13}$C$_{DIC}$ and $\delta^{13}$C$_{CH4}$ values are reported vs. the Vienna Pee Dee Belemnite standard.

Samples for water isotope analysis were collected as unfiltered 20 mL aliquots in a screw-top HDPE bottle. The bottles were subsampled into 1.5 mL vials with septa closures and loaded into the auto-sampler tray of a CDRS instrument (Picarro V 1102-i model). Each sample was injected and measured 6 times using 2.5 μL of water for each injection. Together with the samples, two secondary international standards (USGS 64444 and USGS 67400) and one internal laboratory standard (NTW – Norwich tap water) were measured, each injected 10 times in order to minimise memory effects. Final isotopic compositions were calculated using the calibration line based on the secondary international standards and reported in ‰ units with respect to V-SMOW on the V-SMOW – SLAP scale. The precision error of the measurements was 0.1‰ for $\delta^{18}$O and 0.3‰ for δD.

**2.4    Other data resources**

Unique insights into the sub-permafrost environment in Adventdalen were available to our study on account of the legacy of geological exploration in the region, currently managed by the Store Norske Spitsbergen Kulkompani. This includes many



unpublished borehole logs and other insights into gas accumulation beneath the permafrost in Adventdalen, which were used
for the benefit of the present paper. Furthermore, deep coring, borehole investigation and geophysical surveys have also been
undertaken in the same region as part of the UNIS $CO_2$ Project (Braathen et al., 2012; Huq et al., 2017; Olaussen et al., In
Press), whose published data resources are used below to augment both our own data from the pingo springs, and
unpublished data from the mining reports.

# 3        Results

## 2.1        Sub-permafrost groundwater chemistry inferred from pingo springs

Table 1 shows the geochemistry of all the water samples collected before the melt seasons of 2016 and 2017 from the open
system pingos in Adventdalen. These waters were typically brackish ($Cl^-$ concentrations 390 – 1600 mg $L^{-1}$), largely lacking
in dissolved oxygen (0.00 – 2 mg $L^{-1}$) and $NO_3^-$ ($\leq$ 0.15 mg $L^{-1}$) and with a pH from circum-neutral to alkaline (pH 6.8 –
8.2). Figure 3a shows oxidation-reduction potential (ORP) measurements, indicating that strongly reducing conditions
(negative ORP) existed nearest the coast (typically < -180 mV at Førstehytte and Lagoon Pingos) whilst higher, more
variable values were encountered up-valley (-189 to + 130 mV) at Innerhytte and River Bed Pingos.

| Date | pH | ORP (mV) | O$_2$ | δ$^{18}$O$_{H2O}$ ‰ vsmow | δD$_{H2O}$ ‰ vsmow | Cl | NO$_3$ | SO$_4$ | DIC | Na | K | Mg | Ca |
|---|---|---|---|---|---|---|---|---|---|---|---|---|---|
| **River Bed Pingo** | | | | | | | | | | | | | |
| 16/4/16 | 7.21 | -192 | 0.31 | -14.3 | -102 | 1540 | 0.04 | 40.1 | 2700 | 1910 | 4.99 | 13.8 | 31.6 |
| 12/4/16 | 7.06 | -12.1 | 0.17 | -14.0 | -101 | 1510 | 0.08 | 43.2 | 2770 | 1980 | 4.95 | 13.0 | 30.7 |
| 12/4/15 | 7.61 | -74.9 | 0.68 | -13.9 | -99.8 | 1450 | b.d | 24.3 | 3710 | 2270 | 6.92 | 20.1 | 32.6 |
| **River Bed Pingo (distal)** | | | | | | | | | | | | | |
| 21/4/17 | 7.18 | 134 | 1.1 | -15.4 | -111 | 1560 | b.d | 2880 | 229 | 1020 | 7.89 | 518 | 459 |
| 5/4/17 | 7.32 | -64.1 | 0.0 | -14.9 | -109 | 1520 | b.d | 1950 | 281 | 916 | 6.89 | 353 | 381 |
| 17/3/17 | 8.15 | -25.1 | 0.0 | -15.1 | -109 | 775 | b.d | 3670 | 305 | 563 | 6.30 | 537 | 534 |
| 19/3/17 | 7.22 | 113 | 2.2 | -15.3 | -109 | 780 | b.d | 3510 | 236 | 539 | 5.96 | 495 | 480 |
| **Innerhytte Pingo** | | | | | | | | | | | | | |
| 19/4/17 | 7.16 | -35.4 | 0.77 | -13.7 | -99.2 | 1530 | b.d | b.d | 2000 | 1690 | 4.14 | 12.7 | 19.3 |
| 15/4/17 | 7.11 | -119 | 0.0 | -13.6 | -98.9 | 1490 | b.d | b.d | 2023 | 1680 | 4.13 | 11.7 | 18.4 |
| 17/3/17 | 6.81 | -67.4 | 0.30 | -13.6 | -99.4 | 1520 | b.d | 1.40 | 2043 | 1700 | 4.48 | 12.3 | 18.9 |
| 21/4/16 | 6.89 | -189 | 0.43 | -14.5 | -103 | 1380 | 0.03 | 38.6 | 3930 | 2310 | 3.70 | 20.6 | 40.0 |
| 12/4/16 | 7.07 | -20.7 | 0.23 | -13.5 | -97.6 | 1410 | 0.02 | 14.5 | 3990 | 2330 | 3.54 | 20.2 | 39.6 |
| 22/4/15 | 6.88 | -20.7 | 0.22 | -13.3 | -95.2 | 1490 | b.d | 17.4 | 3870 | 2360 | 5.30 | 17.7 | 28.1 |
| **Førstehytte Pingo** | | | | | | | | | | | | | |
| 19/4/17 | 7.35 | -195 | 1.1 | -14.4 | -107 | 1100 | b.d | 11.3 | 2430 | 1580 | 5.89 | 13.4 | 20.4 |
| 15/4/17 | 7.34 | -180 | 0.0 | -15.0 | -106 | 1130 | b.d | 12.1 | 2390 | 1580 | 7.41 | 15.2 | 20.1 |
| 16/3/17 | 7.35 | -140 | 0.34 | -14.8 | -105 | 1070 | b.d | 15.6 | 2360 | 1540 | 5.43 | 13.7 | 21.1 |
| 21/4/16 | 7.31 | -238 | 0.49 | -15.7 | -110 | 1058 | 0.03 | 48.5 | 4180 | 2190 | 5.54 | 21.6 | 40.7 |
| 12/4/16 | 7.20 | -199 | 0.30 | -14.7 | -105 | 1100 | 0.15 | 63.7 | 4130 | 2210 | 5.76 | 21.5 | 42.0 |
| 9/4/16 | 7.21 | -192 | 0.31 | -14.7 | -105 | 1100 | 0.10 | 59.3 | 3870 | 2110 | 5.69 | 21.6 | 40.9 |
| 10/5/15 | 7.81 | -202 | 0.90 | -15.7 | -111 | 1100 | b.d | 35.0 | 4540 | 2340 | 10.6 | 23.8 | 30.8 |
| 23/4/14 | 7.25 | -212 | 0.60 | -14.4 | -102 | 1100 | b.d | 53.9 | 7560 | 3430 | 12.0 | 25.8 | 39.6 |



| **Lagoon Pingo** | | | | | | | | | | | | | |
|---|---|---|---|---|---|---|---|---|---|---|---|---|---|
| 19/4/17 | 7.9 | -229 | 2.44 | -15.1 | -108 | 392 | b.d | 121 | 3540 | 1560 | 25.4 | 28.1 | 12.9 |
| 15/4/17 | 8.05 | -202 | 0.00 | -15.2 | -108 | 418 | b.d | 128 | 3480 | 1550 | 26.3 | 29.3 | 13.3 |
| 16/3/17 | 7.71 | -181 | 1.71 | -15.5 | -108 | 396 | b.d | 115 | 2340 | 1130 | 19.7 | 21.0 | 9.20 |
| 10/4/16 | 7.94 | -207 | 0.48 | -15.2 | -106 | 541 | 0.05 | 248 | 5250 | 2260 | 39.2 | 63.2 | 38.2 |

**Table 1.** Geochemical characteristics of Adventdalen pingo springs during pre-melt season sampling. All units are in mg $L^{-1}$ unless otherwise stated. $NO_3$ is reported as mg-N $L^{-1}$ and "b.d" means "below detection" (ca. 0.02 mg $L^{-1}$).


With the exception of River Bed Pingo "Distal" in 2017, the generally observed water type was Na-$HCO_3$ with a saturation index (SI) for calcite indicating near-equilibrium ($SI_{calcite} = 0.1 \pm 0.4$) according to WEB-PHREEQ. The dominance of $Na^+$ over the other cations ($Ca^{2+}$, $Mg^{2+}$ and $K^+$: Table 1) and the increasing $Na^+$ to $Cl^-$ ratios towards the coast (Fig. 3a) show how cation exchange (freshening) and rock-weathering effects were increasingly influential down the valley.

Concentrations of $SO_4^{2-}$ in most samples were far lower than expected when compared to late summer baseflow concentrations in local rivers (Hodson et al., 2016). However, at River Bed Pingo a distinct change in spring water chemistry in 2017 to a Mg-Ca-$SO_4$ water type occurred, such that the $SO_4^{2-}$ concentrations increased markedly and the saturation index for gypsum reached equilibrium ($SI_{gypsum} = 0.0 \pm 0.1$) according to WEB-PHREEQ. Outside this period at River Bed Pingo and elsewhere at all times, sub-saturation was observed ($SI_{gypsum} = -2.9 \pm 0.5$). The markedly different Mg-Ca-$SO_4$ water

type suggested the arrival of groundwaters influenced by gypsum-driven de-dolomitization, a process wherein very reactive gypsum catalyses the replacement of dolomite by calcite (Bischoff et al., 1994). However, these samples were collected a greater distance from the pingo than those in 2015 and 2016. Further field observations in 2018 and 2019 (data not shown) clearly suggest there is another groundwater source here and further east. This is further supported by the appearance of waters with different $\delta^{18}O$-$H_2O$ and $\delta D$-$H_2O$ stable isotope characteristics in 2017, which Fig. 3b suggests were more similar

to those encountered at Lagoon Pingo. Therefore, we argue that these samples were either mixtures or rather different waters to those discharging from the pingo.

With the exception of the River Bed Pingo "Distal" waters in 2017, Fig. 3b indicates a general westward depletion (decrease) in both water isotopes towards the coast, where water samples also lie closest to the Local Meteoric Water Line (LMWL) (Rozanski et al., 1993). Although Fig. 4b shows that none of the waters depart significantly from the LMWL, a

linear regression model produces a lower slope (6.09) than that which is associated with the LMWL (i.e. 6.97), suggesting minor isotopic fractionation associated with partial re-freezing (Lacelle, 2011).



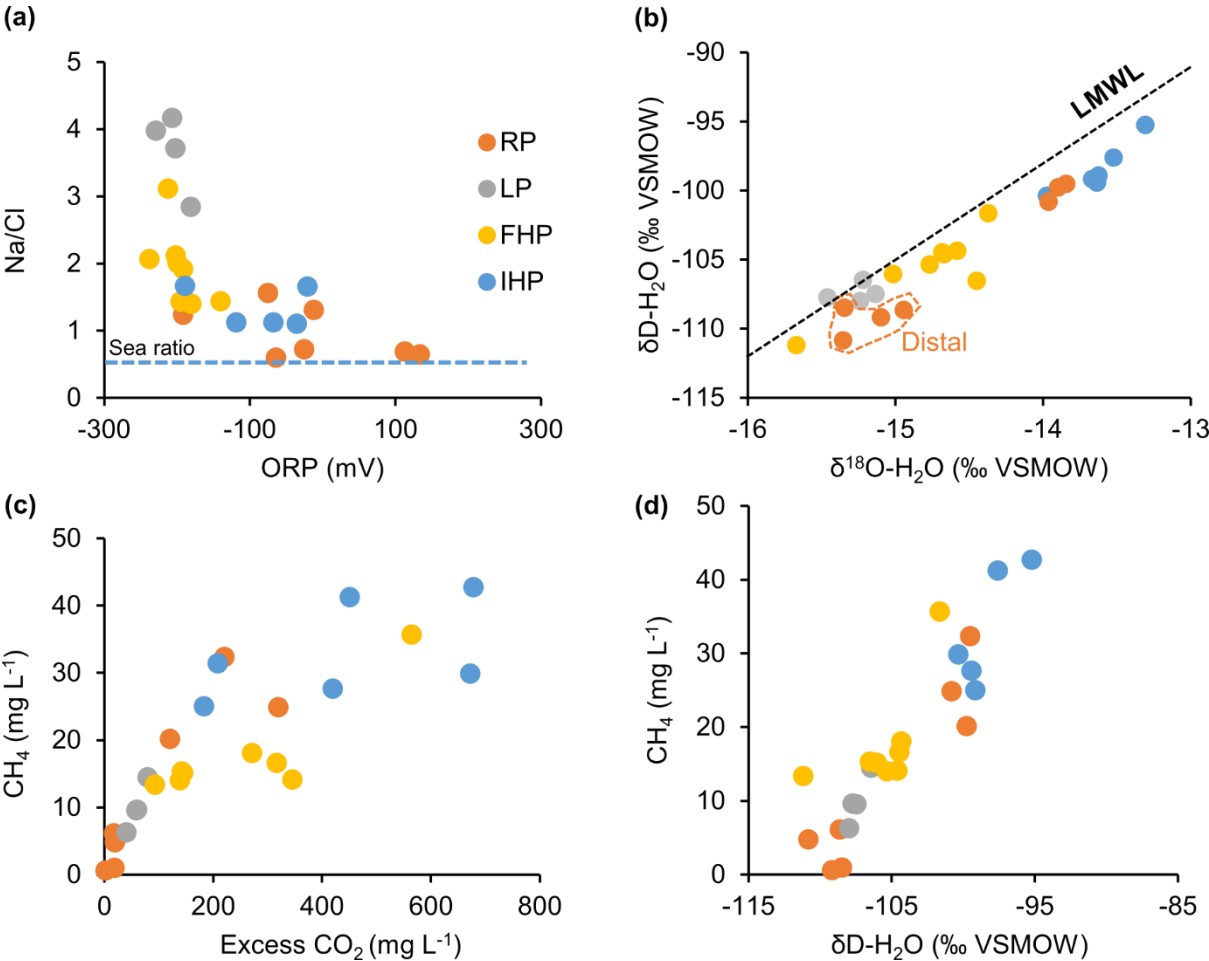

**Figure 3.** Key geochemical and dissolved gas characteristics in spring waters draining River Bed Pingo (RP), Innerhytte Pingo (IHP), Førstehytte Pingo (FHP) and Lagoon Pingo (LP). "LMWL" denotes the Local Mean Water Line. The legend in Fig. 3a applies also to Figs. 3b – 3d.

## 2.2 Sub-permafrost groundwater chemistry inferred from pingo springs

Table 2 shows that concentrations of methane in pingo spring waters in both the pre-melt season and the summer periods lay in the range 0.6 – 42.6 mg L$^{-1}$, which is up to five orders of magnitude greater than calculated atmospheric thermodynamic equilibrium values, and places the most concentrated values marginally above the solubility limit for fresh water at 0 $^{o}$C (i.e. 41 mg L$^{-1}$). The data include samples collected opportunistically from the springs during the summer melt season. The dissolved carbon dioxide concentrations were also in excess of atmospheric equilibrium, by as much as 700 mg L$^{-1}$ at Innerhytte Pingo. Temporal variability in the dissolved gas concentrations was significant at all sites, but greatest at River Bed Pingo, because the arrival of the Mg-Ca-SO$_4$ water type in 2017 was coincident with much lower methane and excess





CO$_2$ concentrations. The variation in methane concentration was positively correlated (p < 0.05) with excess CO$_2$ (r = 0.86:
Fig. 3c), the stable isotopes of water (δ$^{18}$O-H$_2$O, r = 0.86 and δD-H$_2$O, r = 0.91: Fig. 4d) and Na$^+$ (r = 0.74).

| Date | CH$_4$ (mg L$^{-1}$) | eCO$_2$ (mg L$^{-1}$) | δ$^{13}$C-CH$_4$ (‰ VPDB) | δ$^{13}$C-DIC (‰ VPDB) |
|---|---|---|---|---|
| **River Bed Pingo** | | | | |
| 21/4/17 | 4.78 | 19.6 | -54.4 | 12.6 |
| 5/4/17 | 6.23 | 17.3 | -55.0 | 12.5 |
| 17/3/17 | 0.61 | 2.36 | b.d. | 10.1 |
| 19/3/17 | 0.97 | 18.9 | b.d. | 10.1 |
| 16/4/16 | 32.4 | 221 | -55.6 | n.d. |
| 12/4/16 | 24.9 | 320 | -51.5 | n.d. |
| 12/4/15 | 20.2 | 121 | -55.9 | n.d. |
| **Innerhytte Pingo** | | | | |
| 23/9/17 | 25.0 | 183 | -53.8 | 27.1 |
| 19/4/17 | 31.4 | 208 | -55.9 | 26.7 |
| 15/4/17 | 27.6 | 420 | -56.1 | 12.6 |
| 17/3/17 | 30.0 | 672 | -55.7 | 26.3 |
| 21/4/16 | 41.3 | 451 | -57.8 | n.d. |
| 12/4/16 | 42.6 | 678 | -51.8 | n.d. |
| 22/4/15 | 25.0 | 183 | -49.7 | n.d. |
| **Førstehytte Pingo** | | | | |
| 3/10/17 | 11.9 | 641 | -64.2 | 2.5 |
| 13/9/17 | 16.5 | 770 | -64.7 | 2.7 |
| 19/4/17 | 15.3 | 143 | -48.2 | 1.7 |
| 15/4/17 | 15.1 | 145 | -52.3 | 2.4 |
| 16/3/17 | 14.0 | 139 | -54.0 | 2.4 |
| 21/4/16 | 18.1 | 271 | -67.4 | n.d. |
| 12/4/16 | 14.1 | 346 | -55.3 | n.d. |
| 9/4/16 | 16.6 | 317 | -56.1 | n.d. |
| 10/5/15 | 13.4 | 93.0 | -67.1 | n.d. |
| **Lagoon Pingo** | | | | |
| 28/9/17 | 7.26 | 210 | -70.7 | -8.4 |
| 24/8/17 | 9.50 | 58.7 | -69.8 | n.d. |
| 19/4/17 | 6.30 | 40.7 | -55.6 | n.d. |
| 15/4/17 | 9.63 | 60.1 | -48.3 | n.d. |
| 16/3/17 | 13.7 | 79.6 | -62.0 | n.d. |
| 10/4/16 | 9.50 | 58.7 | -66.8 | n.d. |

**Table 2.** δ$^{13}$C composition and concentration of methane and dissolved inorganic carbon (DIC) in pingo springs. "eCO$_2$" is the excess of CO$_2$ relative to equilibrium with the atmosphere. Samples collected opportunistically during the summer are underlined, "n.d." means "not determined, whilst "b.d." means results were below the detection limit.


Table 2 shows that the δ$^{13}$C of methane and dissolved inorganic carbon (δ$^{13}$C-CH$_4$ and δ$^{13}$C-DIC respectively) were variable, especially at Lagoon Pingo and Førstehytte Pingo. The δ$^{13}$C-CH$_4$ lay between -70.7 ‰ and -48.2 ‰ VPDB, which is indicative of biogenic methane at the $^{13}$C-depleted (more negative) end of the scale, and either partially oxidised biogenic or geogenic methane at the $^{13}$C-enriched (more positive) upper end (Schoell, 1980). The data include samples collected





opportunistically from the springs during the summer. Figure 4 shows that the distribution of all measured $\delta^{13}$C-CH$_4$ overlaps closely with that of pore gases extracted from deep cores drilled at the CO$_2$ Well Park south-west of Førstehytte Pingo and Lagoon Pingo (Well B in Fig. 2). These cores penetrated the permafrost, to reveal a pressurized groundwater aquifer just beneath it, and relatively impermeable shale-rich strata further down (see Huq et al., 2017). Methane with $\delta^{13}$C-CH$_4$ between -48.9 ‰ and -52.9 ‰ VPDB, and no other detectable hydrocarbons, is also known to accumulate beneath the

permafrost at Wells C and D in association with a Cl-rich (1500 mg L$^{-1}$) groundwater (Store Norske Spitsbergen Kullkompani, Unpublished Report SN1983-004). The near-absence of hydrocarbons other than methane in the gas just beneath the permafrost at all these sites therefore suggests the accumulation of methane principally from a biogenic source. However, Fig. 4 shows that higher $\delta^{13}$C-CH$_4$ and consistently low ratios of methane to other hydrocarbons (ethane and propane) indicate the presence of a deeper geogenic methane source at depths exceeding 300 m at Well B (Huq et al., 2017).

Adventdalen therefore seems to be characterised by the regional, upward migration of geogenic methane (and other hydrocarbons) at greater depths, prior to its mixing with biogenic methane in the aquifer just beneath the permafrost (Fig. 1b).

In contrast to the $\delta^{13}$C-CH$_4$, the $\delta^{13}$C-DIC values (range -8.5 ‰ to +26 ‰ VPDB) in pingo spring waters did not overlap well with those of the rock pore gases (range -26 ‰ to +21 ‰ VPDB), because they lacked the $^{13}$C depleted

signatures found in the lower part of the core and the permafrost. This cannot be attributed to differences in the DIC speciation among our water samples (containing CO$_{2(aq)}$, H$_2$CO$_3$, HCO$_3^-$ and CO$_3^{2-}$) and the published rock pore gas samples (CO$_{2(g)}$ only). The low $\delta^{13}$C-DIC that is missing from the pingo water samples derives from organic matter respiration and is known to be present in local riverine runoff ($\delta^{13}$C-DIC range -15 ‰ to -4 ‰ VPDB: Hindshaw et al., 2016). However, the higher $\delta^{13}$C-DIC signatures of the pingo springs correspond with those seen in the upper aquifer zone of the cores, and for

$\delta^{13}$C-DIC in excess of +10 ‰ VPDB, may be attributed to the carbon isotope fractionation during the reduction of CO$_2$ to CH$_4$ during hydrogenotrophic methanogensis (Huq et al., 2017; Schoell, 1980).

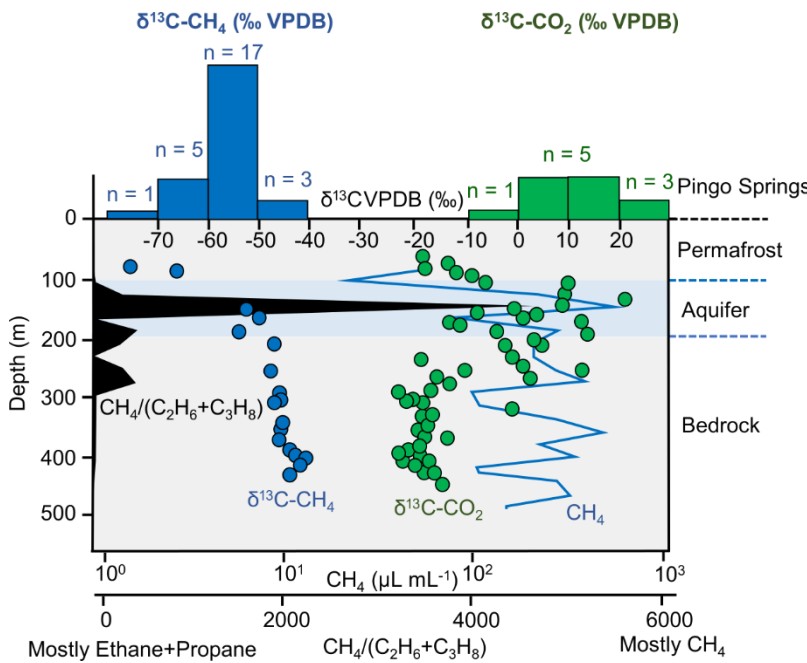

**Figure 4.** Histograms showing stable isotope composition of methane and $CO_2$ in pingo spring waters (from Table 2) for comparison with published pore gases from different depths at the $CO_2$ Well Park (Well D in Fig. 2). The ratio of methane to the sum of ethane and propane (all in $\mu L\ mL^{-1}$) is shown to indicate where biogenic methane is most likely (i.e. high values). Also shown are the approximate lower boundary of the permafrost and the aquifer beneath it.

## 4 Discussion

### 4.1 Groundwater geochemical environment and methane concentrations

The geochemistry of the pingo springs is significantly different to surface waters in the region (Hodson et al., 2016; Rutter et al., 2011; Yde et al., 2008). Their high $Cl^-$ concentrations and distinct $Na-HCO_3$ freshening signature indicate flushing of brackish-marine pore water from the uplifted, Holocene marine sediments of the sub-permafrost aquifer. Importantly, the removal of nitrate and sulphate and the presence of biogenic methane indicate that microbially-mediated processes operate (denitrification, sulfate reduction and methanogenesis, respectively). These decrease the redox potential of the groundwaters towards the low ORP conditions found at the coast (Fig. 3a). The sudden change in the water chemistry at River Bed Pingo during 2017 to one dominated by $Mg-Ca-SO_4$ (Table 1), seems to have resulted from the samples being collected a marginally greater distance away from the pingo. That these "distal" samples contained variable Cl- suggests mixing between groundwaters influenced by the Holocene marine sediments, and another ground water type strongly influenced by gypsum- and dolomite-bearing rocks in the upper part of the valley. The $Mg-Ca-SO_4$ water type also contributed strongly to the higher (yet variable) ORP conditions at the head of the valley and were notable for their lower methane and $CO_2$ concentrations (Table 2).





The strongest predictors of the methane content in the pingo springs are $\delta^{18}O$-$H_2O$ and $\delta D$-$H_2O$ (Fig. 3d). Since the $\delta^{18}O$-$H_2O$ and $\delta D$-$H_2O$ show only a minor departure from the LMWL (Fig. 3b), this indicates a strong water source control upon the gas concentration emerging from the pingos. Methane concentrations generally increase up-valley where $\delta^{18}O$-$H_2O$ and $\delta D$-$H_2O$ become more $^{18}O$ enriched (isotopically higher). The simplest, although initially counter-intuitive explanation

for this change, is an inland increase in the mixing ratio of marine pore waters within the sub-permafrost aquifer. In support of this, a statistically significant ($p < 0.05$) relationship between $Cl^-$ and methane ($r = 0.74$) is apparent when the "distal" samples observed at River Bed Pingo in April 2017 are excluded. The dependence of the methane concentration upon $Cl^-$, $\delta^{18}O$-$H_2O$ and $\delta D$-$H_2O$ is therefore consistent with the expulsion of pore waters from the uplifted marine sediments contributing directly to the high methane concentrations seen in the pingo springs. It is also possible that the cryopegs which

contain unfrozen brines with a high salt content (Keating et al., 2018) enhance the mobility of the groundwaters flowing towards the pingos. These processes are most obvious towards the former marine limit up-valley, because downstream, the system is increasingly diluted by a fresh groundwater with a more depleted (lower) $\delta^{18}O$-$H_2O$ and $\delta D$-$H_2O$ signature that is typical of snow and ice melt from the mountains (Yde et al., 2008).

## 4.2      Methane sources and removal

Comparison of the pingo $\delta^{13}C$-$CH_4$ to the rock core gas samples in Fig. 4 shows that mixtures of biogenic methane (lower $\delta^{13}C$-$CH_4$ signatures) and geogenic methane (higher $\delta^{13}C$-$CH_4$ signatures) might be present beneath the permafrost. However, evidence for a significant contribution from geogenic methane to the pingo springs is equivocal. This is because the partial oxidation of a biogenic methane source may also explain their occasionally higher $\delta^{13}C$-$CH_4$ signatures, due to the preferential oxidation of the $^{12}C$ isotopes (leaving the residual pool $^{13}C$ enriched: Schoell, 1980). For example, the more

variable $\delta^{13}C$-$CH_4$ encountered at Førstehytte and Lagoon Pingos (mean ± 1 standard deviation: -58.8 ± 7.11 ‰ VPDB and -62.2 ± 8.81 ‰ VPDB, respectively) included the only low $\delta^{13}C$-$CH_4$ which can be attributed to biogenic methane with reasonable certainty (Table 2). Rather than invoking rapid switching between two different methane sources, it is far more plausible that this variability is caused by changing degrees of oxidation of biogenic methane during its storage beneath an ice lid. We therefore contend that as storage beneath an ice lid proceeds, the $\delta^{13}C$-$CH_4$ at these sites will increasingly become

greater until hydraulic or thermal fracturing allows the trapped fluids to escape. Refreezing then occurs and the void fills once more with methane-rich fluids with a lower, biogenic $\delta^{13}C$-$CH_4$ signature. As a consequence, the time elapsed since the last fracture event, as well as the volume fraction of the fluids that managed to escape from it before it re-froze, are both likely to cause the notable variations in the $\delta^{13}C$-$CH_4$ of our samples. For this reason, Table 2 shows that samples collected opportunistically at these sites during late summer (when no ice lid existed) consistently showed the low $\delta^{13}C$-$CH_4$ values

(i.e. between -60 ‰ and -70 ‰ VPDB) expected of a biogenic source.

The high methane concentrations at Innerhytte Pingo, observed near the solubility limit (ca. 41 mg $L^{-1}$) on two occasions during 2016, were accompanied by limited variability in $\delta^{13}C$-$CH_4$ (-54.4 ± 2.82 ‰ VPDB). The $\delta^{13}C$-$CH_4$ at nearby River Bed Pingo (-54.5 ± 1.76 ‰ VPDB) was almost identical, and again showed far less variability than at





Førstehytte and Lagoon Pingos. If the high concentrations and consistent $\delta^{13}$C-CH$_4$ are indicative of minimal removal or
carbon isotope fractionation beneath an ice lid, then these results reveal a different (more $^{13}$C-enriched) $\delta^{13}$C-CH$_4$ source
signature than at Lagoon Pingo and Førstehytte Pingo. The higher $\delta^{13}$C-CH$_4$ source signature lies close to the geogenic
methane $\delta^{13}$C-CH$_4$ inferred from the rock cores at depths in excess of 300 m by Huq et al. (2017) (i.e. $\delta^{13}$C-CH$_4$ ca. -45 ‰
VPDB: Fig. 4). However, it is closer to the gas discovered just below the permafrost at Wells C and D (i.e. $\delta^{13}$C-CH$_4$
between -48.9 and -52.9‰ VPDB), which is thought to be entirely biogenic on account of neither ethane nor propane being
present (Huq et al., 2017). Furthermore, the high $\delta^{13}$C-DIC (> 10 ‰) at Innerhytte and River Bed Pingos are strongly
indicative of CO$_2$ reduction by the hydrogenotrophic pathway of biogenic methanogenesis (Schoell, 1980). Therefore, for
quite different reasons, biogenic production and partial oxidation of methane also provides the simplest explanation for the
presence of this gas at high concentrations in pingo outflows further up-valley.

### 4.3    Pingos and springs as methane emission hotspots

In spite of there being evidence for sub-surface oxidation and/or methanotrophy during winter, our data clearly show that
terrestrial seepages formed following isostatic uplift may avoid the effective removal processes associated with methane
seepage from the sea floor. Our results also show that some waters reach dissolved methane concentrations at the highest
attainable levels (i.e. 41 mg L$^{-1}$ in freshwater at 0 $^{O}$C), due to an abundant methane supply and favourable thermodynamic
conditions at the seepage point of the springs. Therefore the potential contribution of the pingos to the annual land-to-
atmosphere methane flux certainly requires appraisal. We note that the equilibrium concentration of dissolved methane for
spring waters remaining in contact with the atmosphere's background CH$_4$ concentration of ~1900 ppb is very low (< 1 x 10$^{-4}$ mg L$^{-1}$) compared to the observed concentrations, and so the emission flux can be crudely estimated from the product of the
pingo outflow rates and their average methane concentration. This is justifiable because outflows are visibly uniform
throughout much of the year, even during winter, when emissions continue following the thermal cracking and hydrofracture
of the ice lid (see Hodson et al., 2019).

Published values of the pingo spring discharges are scarce, but range from 0.01 to 3 L s$^{-1}$, which is similar to our
previous estimates (Liestøl, 1996; Yoshikawa, 1993) and in broad agreement with the numerical modelling of Adventdalen's
groundwater system by Hornum et al. (In Review). This modelling work predicts the likely groundwater discharge
associated with permafrost aggradation in the uplifted fjord sediments throughout the Holocene. It shows that the range of
total groundwater discharge attributable to this process, driven by expansion as the permafrost freezes below the former
Holocene Marine Limit (HML), is between 25.4 and 56.7 m$^3$ day$^{-1}$. Most of this discharges through Lagoon Pingo,
Førstehytte Pingo, or into the sea, because River Pingo and Innerhytte Pingo are closest to the HML. However, our field data
has revealed variations in the mixing ratios of sea water and a fresh, isotopically-depleted water source, which is increasingly
important towards the coast. Therefore, the effects of additional recharge waters from outside the Holocene Marine Limit
upon groundwater discharge need to be considered. This is difficult, but might explain why measured flows at Lagoon Pingo
reported by Hornum et al. (In Review) of 0.26 L s$^{-1}$ are somewhat larger than predicted flows from permafrost aggradation



alone (typically < 0.1 L s⁻¹). Further, another, larger discharge at a neighbouring lake was also discovered whilst this paper was being published, with a flow estimated at twice the Lagoon Pingo spring (0.52 L s⁻¹), thus making the actual outflow at Lagoon Pingo and its environs ~ 0.8 L s⁻¹ (see Table 3). The average methane concentration of the newly discovered lake outflow was 22.2 ± 1.9 mg L⁻¹ according to three samples from February, March and October, 2018. At these times, the electrical conductivity was very similar to the outflow at Lagoon Pingo. The total discharge from the sub-permafrost groundwater system was therefore assumed to be 1.9 L s⁻¹, as in Hornum et al.'s (In Review) scenario that includes additional recharge (Fig. 9, Hornum et al. In Review). However, since this model under-predicts the Lagoon Pingo discharge observations, we conservatively assume that the additional water is accounted for by an overestimate of the discharge into the fjord. This is reasonable on account of the very small hydraulic conductivities in the delta, making it difficult to predict whether the water discharges to the fjord or the pingo. Therefore Table 3 shows a smaller outflow into the fjord (0.22 L s⁻¹) than is presented by Hornum et al. (In Review) (0.89 L s⁻¹).

| Source | Spring discharge (L s⁻¹) | Average CH₄ concentration (Mg L⁻¹) | Annual CH₄ Flux (kg CH₄ yr⁻¹) |
|---|---|---|---|
| River Bed Pingo | 0.11 | 25.8 | 89.6 |
| Innerhytte Pingo | 0.29 | 31.8 | 291 |
| Førstehytte Pingo | 0.46 | 15.0 | 218 |
| Lagoon Pingo | 0.26 | 9.3 | 76.4 |
| Lagoon Lake | 0.52 | 22.2 | 364 |
| Fjord | 0.22 | 20.8 | 145 |
| **Total** | 2.04 | 20.2 | 1183 |
| Atmospheric flux | | | 1039 |
| Active layer emissions | | | 6040 – 10400 |

**Table 3.** Sub-permafrost groundwater discharge and CH₄ flux estimates. The atmospheric flux is the sum of all separate flux except that discharging into the sea ("Fjord"), which is uncertain and assumed to be oxidised. Active layer emissions are median annual fluxes from individual chambers, reported by Pirk et al. (2012).

The methane concentrations that were combined with the above spring discharges to estimate the emission fluxes from each pingo were the average of observations from each pingo shown in Table 2. The potential atmospheric emission for the entire valley then becomes the sum of the individual pingo fluxes. Table 3 shows that the total methane flux via the pingos is ca. 1040 kg CH₄ y⁻¹. Although removal effects have already influenced the concentration values used for the calculations, there is some residual uncertainty due to the inundation of two sites (Lagoon Pingo and River Pingo) during the summer, when samples could not be collected here. At Lagoon Pingo, Hodson et al. (2018) show that a pond forms above the groundwater spring and reduces the annual flux to 42 kg CH₄ yr⁻¹, which is 0.65 times that in Table 3. However, the study of Hodson et al. (2018) ignores diffuse emission sites at Lagoon Pingo associated with a series of water-filled cracks around the pond. Chamber measurements are required to better understand the mode of emission from such micro-sites. At River Pingo, the Advent River inundates much of the floodplain surrounding the pingo during early summer, which most likely reduces emissions because the spring is at its side. However, it is not possible to gain access to this site at such a time





to confirm this, and there are also smaller, more diffuse groundwater emission sources elevated above the river but excluded from our present study. We therefore chose to ignore these uncertainties until the system is better-understood. Similarly, the

temptation to present an upper flux scenario based upon the highest published estimates of pingo spring discharges at Innerhytte and Førstehytte pingos was resisted in favour of a simple sensitivity analysis, which we present below, after a comparison with active layer emissions is undertaken.

Rates of methane emission from chambers installed over the course of three years in an ice wedge polygon site lie in the range $0 - 5$ gC m$^{-2}$ yr$^{-1}$ according to Pirk et al. (2018), with a median of ca. 1 gC m$^{-2}$ yr$^{-1}$. Assuming that all other

wetlands in the valley floor contribute equally (from an area of 4.7 km$^2$), the total active layer emissions are typically 6040 kg CH$_4$ yr$^{-1}$. Our estimates of pingo spring methane emissions amount to 16 % of this flux. The flux from known pingo springs therefore constitutes a significant contribution to landscape CH$_4$ emissions, in spite of their combined discharge of < 2 L s$^{-1}$.

The discovery of methane-rich sub-permafrost groundwaters discharging from Svalbard's open system pingos

means that other perennial springs also deserve attention, because they also be discharge sub-permafrost groundwater. Evidence for this includes the "distal" groundwaters at River Pingo, which are most likely exploiting the faults in the upper valley. Modelling studies also imply that an increase in the discharge of groundwater systems into surface hydrological networks can be expected as climate change proceeds (Bense et al., 2012). Table 2 suggests their methane concentrations might be lower (e.g. $0.5 - 5$ mg L$^{-1}$), although we cannot comment on the efficacy of methane removal or outgassing prior to

sampling. However, since these perennial springs result in the formation of winter icings similar to those encountered on the summit or flanks for the pingos, their detection is greatly facilitated. As a consequence, it is well known in Svalbard that they constitute groundwater flows greatly in excess of those observed flowing from pingos (Bukowska-Jania and Szafraniec, 2005). The sensitivity of the terrestrial atmospheric methane flux is very significant. For example, a combined total groundwater discharge of just 50 L s$^{-1}$ and a methane concentration of 17.9 mg L$^{-1}$ (i.e. mean of all values in Table 2) would

increase Adventdalen's terrestrial methane emissions (i.e. from both wetlands and groundwater) by five times. Such a groundwater discharge would still only represent 0.001 % of the total annual runoff in Adventdalen during the study (A. Nowak and A. Hodson, Unpublished Data). Evidence for similar coastal groundwater springs with high methane concentrations are already known from the MacKenzie Delta, Alaska, where they are thought to constitute approximately 17 % of the emission from the delta (Kohnert et al., 2017). All forms of perennial groundwater discharge in Arctic coastal

lowlands clearly deserve closer attention in order to better understand changes in the release of sub-permafrost methane to the atmosphere.

## 5  Conclusion

The development of open system pingos in Svalbard's coastal lowlands is linked to permafrost aggradation following isostatic uplift. This mechanism results in the expulsion of methane-rich fluids over the course of centuries at individual
sites, and establishes pingos as hotspots for greenhouse gas emissions. In Central Spitsbergen, concentrations of methane (flow weighted average 17.9 mg L$^{-1}$) in four open system pingos greatly exceed those of other, poorly defined groundwater seepages (0.5 – 5 mg L$^{-1}$ average methane concentration). The pingos are therefore important point sources of emissions whose monitoring offers perhaps the best opportunities to gain insights into sub-permafrost methane dynamics. Since this is one of the least understood potential emission sources, open system pingos deserve greater research attention, so that sub-

permafrost emission sources can be integrated with those from the active layer for better emission forecasts. In our study, ca. 1040 kg CH$_4$ yr$^{-1}$ released from just four pingos with a combined fluid discharge of < 2 L s$^{-1}$ increased the landscape emission by ca. 16 %.

## 6        Data Availability

Detailed water quality parameters, including methane concentrations and isotopic composition, for groundwater springs

discharging from open system pingos in Adventdalen, Svalbard (2015-2017) are available at https://doi.org/10.5285/3D82FD3F-884B-47B6-B11C-6C96D66B950D.

## Author Contributions

AJH, AN, PB and MTH collected the samples and analysed the data, with significant input from SJ, KR and AVT. The laboratory samples were analysed by SFT, AJH, KR and AVT. AJH wrote the manuscript, with equal editorial input from

the remaining authors.

## Competing financial interests

The authors declare no competing financial interests.

## Acknowledgements

The authors acknowledge Joint Programming Initiative (JPI-Climate Topic 2: Russian Arctic and Boreal Systems) Award

No. 71126, UK Natural Environment Research Council grant NE/M019829/1, UK Natural Strategic Environment Science Capital Funding, Research Council of Norway grants (NRC nos. 244906 and 294764) and a Royal Geographical Society Ralph Brown Expedition Award 2017. Andrew Fairburn (University of Sheffield) and Stephen Reid (University of Leeds) are thanked for performing the dissolved gas and chemical analysis of the water samples.





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
