# Peer review of "Open system pingos as conduits for highly concentrated methane seepage in Svalbard"

_The Cryosphere, 2020_

## Referee Comment (RC1) · Anonymous Referee #1 · 18 Mar 2020

The Ms "Pingos as hot spots of methane emissions" is a very interesting ms about the groundwater flow related to the special features of pingos. The Ms is very well written, the results are presented and discusses in concise manner. Almost perfect. . . .

Only one major drawback I realized at the end of the study: The emissions or flux of methane from a water body (river, sea, lake) is related to the difference between the measured concentrations (Cw) and the equilibrium concentration (Cequ) of methane in this water; and the gas-water transfer velocity (k). $J = (Cw - Cequil) \times k$ Thus, the calculation and assumptions drawn are wrong, and this part has to be corrected! Also, the way the flux / emission is finally calculated should be explained in the M&M section.

References: Striegl RG, Dornblaser MM, McDonald CP, Rover JR, Stets EG (2012) Carbon dioxide and methane emissions from the Yukon River system. Glob Biogeochem Cycl 26 Wanninkhof R (2014) Relationship between wind speed and gas exchange over the ocean revisited. Limnol Oceanogr: Methods 12:351-362

---

## Author Comment (AC1) · 30 Mar 2020

We thank the author of these comments and note that they raise an interesting point about the diffusion-based approach to flux calculation. Below we address the reason we did not employ such an approach in our paper. However, before we do so, we also point out that we have already used it for one of our sites where the local conditions make this approach entirely appropriate (i.e. Lagoon Pingo: See Hodson et al, 2018 in the reference list).

First, our manuscript acknowledges that it "crudely" estimates the methane release from pingo waters to the atmosphere. It does so by assuming that all waters achieve equilibrium with the atmosphere due to turbulence and freezing effects. These are

discussed further below because both these effects render the diffusion modelling approach unsuitable.

Secondly, the more sophisticated diffusion approach is almost impossible to employ, because it requires environmental data (wind speed, temperature etc) that are extremely variable.

Therefore, in defense of our approach: A) Pingo waters are not similar to open ocean, shallow lakes or river waters, because: i) Often pingo springs are shallow (<5cm deep), turbulent and sometimes gas-saturated flows that are unlikely to behave like deeper surface waters in lakes and the ocean surface. ii) Seasonality strongly affects water bodies of springs (extent and depth of water body). In summer, some of the pingos develop shallow pools and slow moving springs where the diffusion modelling approach is certainly appropriate. This applies to the two sites at Lagoon Pingo for example (hence our earlier manuscript in Frontiers). However, for the rest of the year, the flows discharge from cracks in an ice lid and form an icing that is constantly changing. iii) Freezing and hydrostatic pressure forces pingo waters through the ice lid and into the atmosphere. Sudden, turbulent drainage events occur throughout much of the year (say 8 – 9 months). These turbulent flows conditions are not at all similar to those where the Wanninkhof equations that underpin the diffusion approach may be applied iv) Ice formed in methane rich waters during freezing is generally quite reduced in methane (Langer et al 2015, Biogeosciences, 12, 977–990, doi:10.5194/bg-12-977-2015). In other words, exclusion of the dissolved gas from the ice during freezing increases the concentration of methane in the residual water and thus enhances the emission into the atmosphere. This is not east to accommodate into the diffusion modelling approach

B) Not all pingo springs have been identified or their flows, or methane content, properly quantified. High exposure environments near pingo summits (high wind speeds) means that predicted values are likely to be substantially underestimated since wind speeds are not well known for these sites (See Wanninkhof 2014)

Interactive
comment
C) Supersaturated waters may not behave as predicted from sub-saturated ocean waters. In some cases (e.g. Innerhytte Pingo) ebullition is the dominant process at times (ie not diffusion), which is why the samples exceed the solubility limit here. A more appropriate flux method here could involve the use of chambers, however, these would be extremely difficult to employ, and add a further methodology to hamper comparison.

D) Turbulent flow from pingo springs are likely to mean that diffusion-based approach is not appropriate for this system.

We would be happy to improve our paper by discussing these issues and regret not making the methods and their justification clearer.

---

## Referee Comment (RC2) · Anonymous Referee #2 · 5 May 2020

The manuscript 'Open system pingos as hotspots for sub-permafrost methane emission in Svalbard' by Hodson and co-workers report concentrations of dissolved ions and gases as well as water and methane stable isotope signatures in spring water of four pingos in the Adventdalen valley, Spitsbergen. The springs were sampled several times between 2014 and 2017. The authors use the obtained data to derive information on the sources of the sampled ground water and of the detected methane, which is present in surprisingly high concentrations. Furthermore, they estimate the contribution of the spring water methane to total atmospheric methane fluxes from the Adventdalen valley.

The topic of the paper is of high relevance and I read it with great interest. It's concise and well written, but it could be improved by adding some background information.

Furthermore, the discussion is in large parts very speculative due to the lack of data.

The development of an open system pingo should be explained in more detail. It was unclear to me, how liquid water may find it's way through the permafrost. What is the temperature of the permafrost? Are these open system pingos particularly developing above marine sediments? What is the difference to a 'normal' pingo? It's particular difficult to understand since the cited reference (L57) is not given in the list of references.

Furthermore, more background information should be given on the geology of the study sites, including the geology of the surrounding mountains that may affect the composition of the spring water. Is there a connection between the springs and fresh melt water (as suggested in lines 275ff)? Is there a talik below the river and might this be connected to the springs? Furthermore, more background information on the potential source of the methane in the spring water should be given. The authors differentiate biogenic and geogenic sources. However, it should be made more clear which geogenic sources might be present, gas hydrates or natural gas from deep deposits? Is there information on these sources in the region, and what is the carbon stable isotope signature of these sources? Concerning the biogenic source, it should be explained why high methane concentrations in marine waters are expected. Generally, no methane is produced as long as sulfate is present. In contrast, methane, e.g. from gas hydrates is oxidized with sulfate as electron acceptor.

The discussion concerning the methane sources is rather speculative due to a lack of data. The carbon stable isotope signature is only a weak indicator for differentiating geogenic and biogenic methane sources. If only the carbon stable isotope signatures of methane are available and no delta D or concentrations of further hydrocarbons, as in this manuscript, no differentiation between geogenic and biogenic sources is possible. E.g., gas hydrates may have carbon stable isotope signatures between about -40 and -70‰ a range covering the whole values given in this manuscript.

But the weakest part of the discussion is the part on the pingos as methane emission

hotspots. The authors derive spring water fluxes from an unpublished study on Adventdalen's groundwater system, add unpublished data on methane concentrations in a 'neighboring lake', which contributes about 1/3 to the total flux estimate and assume that 100% of the methane in the water will be emitted to the atmosphere. Estimating methane fluxes from water concentrations comes along with high uncertainties. It might be possible for pond, lake or sea water. However, in soils, bacteria will likely oxidize a large fraction of the methane as soon as oxygen is available. Hence, methane fluxes will likely be much lower. To derive meaningful data on methane fluxes from soil surfaces, emission measurements should be conducted. Furthermore, there seems to be a mistake in the calculation of the land fluxes from Adventdalen valley using the Pirk et al. (2017) paper and the active layer fluxes seems twice as high (see specific comments) as given in this manuscript. In this case the relative contribution of the sub-permafrost fluids is reduced by 50%.

Concluding, I suggest changing the title of the manuscripts, since it indeed does not measure methane emissions. Furthermore, I would downplay the calculations of methane emissions and more clearly consider their uncertainties. The authors mention that it is only a 'crude' estimate, which is correct. In this case, this crude estimate should not be in the focus of the manuscript by mentioning it in the title and elaborating it over more than half of the discussion. The authors may dicuss the emissions in a more qualitative way and also include information about the abundance of such springs, if available.

Finally, the reference list needs attention.

Specific comments:

L31: This quote does not fit here very well, better cite particular studies that are 'quantifying the release of methane from the active layer during summer thaw' and not a general review on the permafrost carbon feedback.

L57: This reference is not given in the list of references

L138 ff: How were gas pressures measured in the vials and which CH4 solubility was assumed?

L152: Please give the standards used for methane and CO2 $\delta$13C measurements.

L214: Excess CO2 seems to correlate with methane concentration not it's variation.

L225f: What means 'overlaps closely'?

L239ff: The last part of this paragraph belongs to the discussion.

L258ff: The 'distal' samples not only seem to be different but they very obviously are.

L265ff: I find this paragraph confusing and the conclusions not convincing. It is indeed counterintuitive to expect that the influence of marine waters are higher the farther one comes from the sea. Furthermore, this conclusion is only supported if a part of the dataset (distal samples) is omitted from the analysis, but there is no justification given to do so. Furthermore, it is unclear why the system is more diluted downstream by fresh groundwater from snow and ice melt. I understand, also from Fig. 4 that the sampled water originates from below the permafrost. In this case, the up-valley sites should be more influenced by melt water.

L286ff: The explanation of the variability in CH4 stable isotopes is unclear. Why should CH4 oxidation preferentially take place while the fluids are trapped below an ice lid and not during it's transport to the surface or after surface thaw? To oxidize methane, an electron acceptor is needed, the respective microorganisms and liquid water but not stagnant water. And what might be the electron acceptor for methane oxidation? The fluids seem mostly oxygen free and low in sulfate.

L310: What means 'favorable thermodynamic conditions' in this context? Favorable for which process?

L317ff: I understood from L286ff that the springs are frozen in winter. Please clarify.

L 331FF. Please give the reference for 'this paper'. Furthermore, clarify to which paper

the newly introduced data from the 'neighboring lake' belong.

L345: Pirk et al., 2017?

L347f: This calculation neglects aerobic methane oxidation, which might oxidize up to 100% of the methane before it is released into the atmosphere. Hence, the flux assumption from the springs is the upper limit of methane fluxes from the springs.

L363: Pirk et al. (2017 not 2018) report 'typically . . . a . . . seasonal budget of around 2 gC m-2' (not 1 g C m-2) for the summer thaw season (1st June to 30th September) in Adventdalen. According to my calculation the annual flux from 4.7 km2 would then be about 12,600 kg methane yr-1 (not 6,040 kg methane yr-1). Furthermore, the winter fluxes are not considered in these estimates, which might be as high as the summer fluxes (see Zona et al., 2016).

L375f: The meaning of this sentence ('The sensitivity. . .') is unclear.

L376ff: I cannot follow this calculation. Where does the number 50 L sec-1 come from? What is the Adventdalen terrestrial methane flux? In addition, why compare the total annual runoff of Adventdalen with the groundwater flux of 50 L- sec-1? The authors are aware that the methane concentration in surface melt water is several orders of magnitude lower than what they found in the springs with sub-permafrost fluids. This comparison is without meaning.

Table 2: Please also differentiate the "distal" samples from the River Bed Pingo

Fig. 2 is difficult to read. Please give references for the published pore water data and please use units that make a comparison with the data in the tables possible (e.g. mg L-1 not $\mu$L mL-1)

Cited references:

Pirk N., Mastepanov M., Loez-Blanco E., Christensen L.H., Christiansen H.H., Hansen B.U., Lund M., Parmentier F.J.W. et al. (2017) Toward a statistical description of

methane emissions from arctic wetlands. Ambio, 46, S70-S80.

Zona D., Gioli B., Commane R., Lindaas J., Wofsy S.C., Miller C.E., Dinardo S.J., Dengel S. et al. (2016) Cold season emissions dominate the Arctic tundra methane budget. Proceedings of the National Academy of Sciences of the United States of America, 113, 40-45.
* * *

---

## Author Comment (AC2) · 6 May 2020

We thank the reviewer for the comments and agree with some, but not all of the criticisms raised. There are also some good suggestions made that could improve the paper. Here we reply to the general comments raised, rather than the specific ones.

The referee comments about the lack of data and the speculation that materialises in the Discussion. We wish to point out that there is a necessity to speculate to some degree on account of the great lack of studies pertaining to fluid flows beneath continuous permafrost. van der Ploeg (2012) describe it as "the fierce data scarcity of subpermafrost groundwater systems". We are therefore describing an important, yet largely unknown fluid migration pathway. Our system is in fact data-rich by comparison

to other sub-permafrost environments, but we still lack the kind of access enjoyed by researchers studying methane dynamics in wetlands, the active layer or other near-surface environments.

The referee asks for a better description of the formation of open system pingos and also the hydrogeology of the valley. We point out that these processes form the basis of Hornum et al's submission to The Cryosphere, presently available as a Discussion paper (Hornum et al, 2020) and cited in the field site section of our paper (line 82). The characteristics and relationships between isostatic uplift, fluid flow and the geology are discussed at length by Hornum et al, and we can therefore make the links between these papers better for the reader.

We regret the mistake with the reference given on line 57. It is given as Hodson et al (2018) and should in fact be Hodson et al (2019), as given in the reference list. The paper describes open system pingo formation at length.

There is no talik beneath the river. It is a good suggestion to include this information in a revised version of the paper.

More background on biogenic and geogenic methane is requested and it is asked whether the geogenic methane is likely to be hydrate-derived or natural gas from depth. We are able to comment upon this given our earlier work on the thermodynamic stability of gas hydrates in the region (Betlem et al, 2018). This clearly show that hydrates are no longer stable in the valley bottoms, yet they might persist in the mountains. However, the issue opens a can of worms, because the rate of hydrate dissolution and the fate of the gas produced is unclear. Therefore, since hydrate is not a source of CH4 but a store, we do not see the need to write much about this. The issue is covered to some extent by Hornum et al (2020) though, and we can allude to this and Betlem et al (2018) if deemed necessary.

The reviewer asks whether there is information on the geogenic sources, which we are happy to enhance following a recent publication. However, we already discuss Huq

et al (2017) on p11 and 12, who makes it clear that: i) a mixture of both biogenic and geogenic methane exists beneath the permafrost, but ii) only biogenic methane is found immediately beneath the permafrost in the aquifer that provides runoff to the pingos.

The reviewer raises a good point about our assumed links between biogenic methane and pore water flow through marine sediments being unlikely on account of there usually being abundant SO4 in these sediments to enable sulphate reduction to out-compete methanogenesis. We made the point because Cl correlates with methane concentrations. We now think other processes could cause this and so we can make the necessary changes.

The reviewer implies that our discussion about methane provenance is weak because it is entirely based upon 13C-CH4. The reviewer then implies that we cannot discriminate between geogenic methane, biogenic methane and hydrates. But hydrates are not a source of methane – they are a transient store of either geogenic or biogenic methane (or a mixture of the two). Our paper therefore seeks to present the following logical narrative that we believe is entirely robust:

a) The source is either geogenic or biogenic. It doesn't matter if it was from a hydrate or not (although this is clearly an interesting question that we are able to address through modelling)

b) The 13C-CH4 can rule out geogenic when the values are low, and we have many low values that fall outside this "geogenic range".

c) An earlier study of methane in pore spaces conducted at our site uses CH4 concentrations, 13C-CH4 and the presence of other hydrocarbons to establish the relative abundance of biogenic versus geogenic methane from the surface down to ca. 900m (Huq et al, 2017). Our figure 4 presents these data down to 500 m. This work does not require delta-D because the presence of other hydrocarbons is a reliable indicator of geogenic CH4 in the region (Huq et al, 2017).

d) The above study shows that there is a methane source in an aquifer immediately below the permafrost that is largely biogenic. Geogenic methane is found at greater depths (> 300 m: see Fig 4) and might not be able to migrate upwards due to the geology of the site

e) The biogenic methane inferred from Huq's study was also found by a mining company, who reported a salty groundwater body with a 13C-CH4 range (-48.9 ‰ to -52.9‰ almost identical to that found at our nearby pingos sites (ie River Pingo and Innerhytte Pingo: -49.7 ‰ to -57.8‰ as in Table 2). Their reported salt content was 1500 mg/L, which is also almost identical to that found at these two pingos (1380 – 1540 mg/L: Table 1).

f) At the other two pingo sites, the 13C-CH4 values either lie within the same range as the above, or are too low to be geogenic.

g) We therefore conclude that there is no evidence for geogenic methane in our springs.

The reviewer suggests that the weakest part of our manuscripts is where we explore the emission fluxes. We tend to agree but feel that their potential significance should still be addressed. We don't mind attempting it qualitatively, as proposed. We thought about this earlier, but then decided that a quantitative argument would be a reasonable demand from any likely reviewer. However, we do not agree with some of the criticisms directed towards our emission estimates, and wish to make the following points in defence of our work:

i) Methane consumption in soils "will likely oxidize a large fraction of the methane as soon as oxygen is available". We point out that soils are frozen for much of the year, yet the springs we study are constantly discharging. Furthermore, we seldom find the springs infiltrating into soils. More often, the springs erode turbulent channels through impermeable marine clays. Such flows are more conducive to rapid degassing to the atmosphere and therefore little methanotrophy. However, at pingos where lakes form upon their summit then methanotrophy is more likely. The impact of this process is

already discussed (line 310 onwards) and accounted for in our estimates.

ii) "To derive meaningful data on methane fluxes from soil surfaces, emission measurements should be conducted." We do not wish to derive such data. We are studying point sources of methane that by-pass the soil environment. Maybe we could include photographs of the sites to clarify this?

iii) "The authors derive spring water fluxes from an unpublished study on Adventdalen's groundwater system..." This study is Hornum et al, as cited and therefore available to the reader as a discussion paper. It provides a lot of necessary back ground data, as requested above.

iv) "Furthermore, there seems to be a mistake in the calculation of the land fluxes from Adventdalen valley using the Pirk et al. (2017) paper and the active layer fluxes seems twice as high (see specific comments) as given in this manuscript." The reviewer continues in their specific comments with: "typically...a...seasonal budget of around 2gC m-2 for the summer thaw season..." This is a quote from the Pirk et al paper. Unfortunately, the quote refers to the median of two sites: Adventdalen and Zackenberg (in Greenland). The emissions from Zackenberg are greater than those from Adventdalen. Quick scrutiny of Figure 5 in Pirk et al (2017) clearly shows that all of the median values at Adventdalen lie below 2. It is therefore hard to justify using 2 gC m-2 y-1 as a spatially representative value. For this reason, I digitised Figure 5 in Pirk et al and determined the minimum and maximum median values from their three year study. These values were used to produce the range of likely emissions from wetlands for comparison with our emission estimates from springs. This range is presented in Table 3.

v) "Furthermore, the winter fluxes are not considered in these estimates, which might be as high as the summer fluxes (see Zona et al., 2016)." The Pirk et al (2017) study does infact include the freeze-up processes that were emphasised by the Zona et al (2016) study. After this period, the (late) winter emissions in Adventdalen have not

been studied much, although Pirk et al (2016) did some pre-melt chamber work one April/May and found great suppression of the methane flux by icing layers. Where such layers were less prolific (in Zackenberg again), the winter fluxes were one to two orders of magnitude lower that those before the end of freeze up.

References Betlem, P., Senger, K. and Hodson, A., 2019. 3D thermobaric modelling of the gas hydrate stability zone onshore central Spitsbergen, Arctic Norway. Marine and Petroleum Geology, 100, pp.246-262. Hornum, M.T., Hodson, A.J., Jessen, S., Bense, V. and Senger, K., Numerical modelling of permafrost spring discharge and open-system pingo formation induced by basal permafrost aggradation. The Cryosphere Discussions, https://doi.org/10.5194/tc-2020-7, 2020 Pirk, N., Tamstorf, M.P., Lund, M., Mastepanov, M., Pedersen, S.H., Mylius, M.R., Parmentier, F.J.W., Christiansen, H.H. and Christensen, T.R., 2016. Snowpack fluxes of methane and carbon dioxide from high Arctic tundra. Journal of Geophysical Research: Biogeosciences, 121(11), pp.2886-2900. van der Ploeg, M.J., Haldorsen, S., Leijnse, A. and Heim, M., 2012. Subpermafrost groundwater systems: Dealing with virtual reality while having virtually no data. Journal of hydrology, 475, pp.42-52.

---

## Author Response (AR1)

**Reviewers' comments are italicised, our response is not. Our direct actions are highlighted in red text. Line numbers refer to the marked-up version of the manuscript**

**Reviewer 1.**

*The Ms "Pingos as hot spots of methane emissions" is a very interesting ms about the groundwater flow related to the special features of pingos. The Ms is very well written, the results are presented and discusses in concise manner. Almost perfect.... Only one major drawback I realized at the end of the study: The emissions or flux of methane from a water body (river, sea, lake) is related to the difference between the measured concentrations (Cw) and the equilibrium concentration (Cequ) of methane in this water; and the gas-water transfer velocity (k). J = (Cw - Cequil) x k Thus, the calculation and assumptions drawn are wrong, and this part has to be corrected! Also, the way the flux / emission is finally calculated should be explained in the M&M section.*

We appreciate the positive comments and acknowledge that the perceived drawback related to our flux calculations needs to be addressed in the paper. Since no comments other than the flux calculation issue require consideration, we deal only with this point below.

Our manuscript acknowledged that it "crudely" estimates the methane release from pingo waters to the atmosphere. It did so by assuming that all waters achieve equilibrium with the atmosphere due to turbulence and freezing effects. This is further simplified by the fact that the equilibrium concentration is negligible compared to the initial concentration and so can be ignored. The equation given by the reviewer is most relevant for cases where equilibrium is not achieved – for example with standing water bodies like lakes, or the sea, with a continuous influx and significant residence time caused by storage. However, envisage a turbulent spring flowing without such storage in a pond, and freezing while it does so. These conditions render the above calculations rather unsuitable, and they fail because the coefficient *k* is impossible to define. A further issue is that the proposed method does not account for the ebullition of gas, and so might underestimate the gas flux. The reviewer's comment is valid for one or two of our sites during the summer though, and we completely accept this criticism. In fact we have already employed the recommended approach in a different paper about one of these sites (Hodson et al, 2019). Given these uncertainties, we took the Editor's advice and presented a more qualitative argument about the likely importance of the fluxes to the atmosphere. Further details are below – but note the extensive changes to Section 4.3.

**Reviewer 2.**

We thank the reviewer for also complementing the paper for being and well-written.

*The development of an open system pingo should be explained in more detail. It was unclear to me, how liquid water may find it's way through the permafrost. What is the temperature of the permafrost? Are these open system pingos particularly*

*developing above marine sediments? What is the difference to a 'normal' pingo? It's particular difficult to understand since the cited reference (L57) is not given in the list of references.*

The citation, which describes the formation process for lagoon Pingo has been corrected (line 58). Since only a brief description about formation is given in the introduction (lines 58 – 61), more comments on the Adventdalen pingos are included in Section 2.1 (lines 73 – 81), where there are three further citations. Unfortunately, we do not know the temperature of the permafrost in this area.

*Furthermore, more background information should be given on the geology of the study sites, including the geology of the surrounding mountains that may affect the composition of the spring water. Is there a connection between the springs and fresh meltwater (as suggested in lines 275ff)? Is there a talik below the river and might this be connected to the springs?*

We acknowledge the need for more background information regarding the genesis of the pingos, the geology and the type of methane that might be present. This information is now included in the field site section, in lines 95 onwards. We also improved the representation of the geology in Figures 1 included a new Figure 2b – all described from lines 95 to 108. Two new citations have been included too. From lines 110 to 117, we have used new citations and added additional text (some of which was originally described in "Section 2.4: Other data resources") to better integrate existing knowledge into the description of the field site. The discussion also uses this information to explore the links between the shale unit and the geochemistry, due to the unusual observations at River Bed Pingo (distal site) in 2017 (line 297 onwards).

*Furthermore, more background information on the potential source of the methane in the spring water should be given. The authors differentiate biogenic and geogenic sources. However, it should be made more clear which ge-ogenic sources might be present, gas hydrates or natural gas from deep deposits? Is there information on these sources in the region, and what is the carbon stable isotope signature of these sources?*

We have included more information in the above section between lines 110 – 117. Since hydrates could be composed of any gas origin (ie bio or geo-genic) and since it is not known if any are indeed present in the valley, we decided not to discuss hydrates in the introduction.

*Concerning the biogenic source, it should be explained why high methane concentrations in marine waters are expected. Generally, no methane is produced as long as sulfate is present. In contrast, methane, e.g. from gas hydrates is oxidized with sulfate as electron acceptor.*

The reviewer raises a good point here on account of the potential for sulphate reduction to out-compete methanogenesis. We made the point because Cl correlates with methane concentrations. We now think other processes could cause this and so we have removed this point.

*The discussion concerning the methane sources is rather speculative due to a lack of data. The carbon stable isotope signature is only a weak indicator for differentiating geogenic and biogenic methane sources. If only the carbon stable isotope signatures of methane are available and no delta D or concentrations of further hydrocarbons, as in this manuscript, no differentiation between geogenic and biogenic sources is possible. E.g., gas hydrates may have carbon stable isotope signatures between about -40 and-70‰ a range covering the whole values given in this manuscript.*

The reviewer implies that our discussion about methane provenance is weak because it is entirely based upon 13C-CH4. The reviewer then implies that we cannot discriminate between geogenic methane, biogenic methane and hydrates. But hydrates are not a source of methane – they are a transient store of either geogenic or biogenic methane (or a mixture of the two). We therefore seek only to assess whether there is any evidence for geogenic methane in the water, largely because previous work has demonstrated a clear dominance of biogenic methane just beneath the permafrost at our field site. As a result, we respectfully suggest that there is no need for the dD-CH4 isotopes because:

i) 13C-CH4 alone can rule out geogenic when the values are low, and we have many low values that fall outside this "geogenic range".

ii) An earlier study of methane in pore spaces conducted at our site uses CH4 concentrations, $\delta^{13}$C-CH4, $\delta^{13}$C-DIC and the presence of other hydrocarbons to establish the relative abundance of biogenic versus geogenic methane from the surface down to ca. 900m (Huq et al, 2017). This work clearly shows that the geogenic methane fails to migrate effectively into the aquifer beneath the permafrost. This work does not require delta-D because the presence of other hydrocarbons is used as a reliable indicator of geogenic CH4 instead (Huq et al, 2017). This information is included in Figure 4.

iii) The above study shows that there is a methane source in an aquifer immediately below the permafrost that is largely biogenic.

iv) The biogenic methane inferred from Huq's study was also found by a mining company, who reported a salty groundwater body just beneath the permafrost with a 13C-CH4 range (-48.9 ‰ to -52.9‰) that is almost identical to that found at our nearby pingos sites (ie River Pingo and Innerhytte Pingo: -49.7 ‰ to -57.8‰ as in Table 2). Their reported salt content was 1500 mg/L, which is also almost identical to that found at these two pingos (1380 –1540 mg/L: Table 1).

v) At the other two pingo sites, the 13C-CH4 values either lie within the same range as the above, or are even lower (more 13-C depleted) and therefore too low to be geogenic.

We therefore conclude that there is almost no evidence for a significant geogenic methane contribution to our springs. We have edited lines 250 - 261 and Figure 4 to make incorporation of these other data clearer and more compelling. We also included reference to our new publication about the deeper geogenic gas (Ohm et al, 2019).

*But the weakest part of the discussion is the part on the pingos as methane emission hotspots. The authors derive spring water fluxes from an unpublished study on Adventdalen's groundwater system, add unpublished data on methane concentrations in a 'neighboring lake', which contributes about 1/3 to the total flux estimate and assume that 100% of the methane in the water will be emitted to the atmosphere. Estimating methane fluxes from water concentrations comes along with high uncertainties. It might be possible for pond, lake or sea water. However, in soils, bacteria will likely oxidize a large fraction of the methane as soon as oxygen is available. Hence, methane fluxes will likely be much lower. To derive meaningful data on methane fluxes from soil surfaces, emission measurements should be conducted.*

We tend to agree that the emission estimates are the weakest part of the paper, but feel that their potential significance should still be addressed. We have therefore followed both the reviewer's and the editor's suggestion to achieve this with a more qualitative approach, which is now described between lines 344 – 348.

However, since we respectfully disagree with some of the criticisms directed towards our emission estimates, we first wish to offer the following explanation (that could have been clearer in our initial manuscript):

*Methane consumption in soils "will likely oxidize a large fraction of the methane as soon as oxygen is available".*

We point out that soils are frozen for much of the year, yet the springs we study are constantly discharging, usually over a smooth ice surface. Furthermore, we seldom find the springs infiltrating into soils before much of their methane has been lost. In summer, the springs erode turbulent channels through impermeable marine clays or cascade down the flank of the pingo – which is also conducive to rapid degassing to the atmosphere but not really to methanotrophy. However, at pingos where lakes form upon their summit, then methanotrophy is indeed more likely. These points have been incorporated into the new Section 4.3 along with some new empirical evidence and a new figure (Figure 5) (see lines 371 - 404) that demonstrates rapid methane loss.

*To derive meaningful data on methane fluxes from soil surfaces, emission measurements should be conducted.*

We do not wish to derive such data because we are describing emission from springs, not soils. Our springs by-pass the soil environment. We hope that the new text described above helps clarify this

*The authors derive spring water fluxes from an unpublished study on Adventdalen's groundwater system...*

This study is Hornum et al, currently under revision and available to the reader as a discussion paper. It provides a lot of necessary background data, but we have greatly reduced the dependence upon this paper (at the Editor's request) and have made the whole section simpler and easier to follow. All speculation about springs not sampled in our study has been removed. One additional site, "Lagoon Lake" has been included though, because it is clearly part of the Lagoon Pingo system, and we present sufficient measurements from this site to justify its inclusion (lines 360 – 362). Table 3 has been changed and the source of the information used for the fluxes made clear in the caption.

*Furthermore, there seems to be a mistake in the calculation of the land fluxes from Adventdalen valley using the Pirk et al. (2017) paper and the active layer fluxes seems twice as high (see specific comments) as given in this manuscript. In this case the relative contribution of the sub-permafrost fluids is reduced by 50%.*

We regret that the reviewer has made a mistake – which is easy to do on account of the wording in the Pirk paper. This issue is dealt with under specific comments below.

*Furthermore, the winter fluxes are not considered in these estimates, which might be as high as the summer fluxes (see Zona et al., 2016).*

The Pirk et al (2017) study does in fact include the freeze-up processes that were emphasised by Zona et al's (2016) study. After this period, the (late) winter emissions in Adventdalen have not been studied much, although Pirk et al (2016) did some pre-melt chamber work one April/May and found great suppression of the methane flux by icing layers. Where such layers were less prolific (in Zackenberg, Greenland – not Svalbard), the winter fluxes were one to two orders of magnitude lower that those before the end of freeze up. Therefore, no amendments to the manuscript were deemed necessary concerning this point.

*Concluding, I suggest changing the title of the manuscripts, since it indeed does not measure methane emissions. Furthermore, I would downplay the calculations of methane emissions and more clearly consider their uncertainties. The authors mention that it is only a 'crude' estimate, which is correct. In this case, this crude estimate should not be in the focus of the manuscript by mentioning it in the title and elaborating it over more than half of the discussion. The authors may dicuss the emissions in a more qualitative way and also include information about the abundance of such springs, if available.*

We have changed the title to better reflect the role of the pingos and have also changed the signposting at the end of the introduction section to emphasise that the paper is largely about the exploitation of pingos by the gas-rich fluids (line 62 to 70). We have also changed the emphasis of Section 4.2 to remove emission estimates.

*Finally, the reference list needs attention.*

Done

*Specific comments:*

*L31: This quote does not fit here very well, better cite particular studies that are 'quantifying the release of methane from the active layer during summer thaw' and not a general review on the permafrost carbon feedback.*

A new citation has been added that incorporates active layer emissions and methane cycling into a global review. I changed the sentence to make the lack of explicit active layer studies acceptable.

*L57: This reference is not given in the list of references*

The citation has been corrected (see line 58)

*L138 ff: How were gas pressures measured in the vials and which CH4 solubility was assumed?*

Details of the number of standards, the linear calibration range used and the detection limit are now included lines 171 - 174. The analytical process does not measure pressure as no pressurisation occurs during headspace formation. Appropriate amendments have been made to the text to make this clear (line 169). We use the Bunsen Solubility Coefficient (a proxy for gas solubility) to account for temperature effects upon the solubility value.

*L214: Excess CO2 seems to correlate with methane concentration not it's variation.*

Changed (line 239)

*L225f: What means 'overlaps closely'?*

Changed (on line 250)

*L239ff: The last part of this paragraph belongs to the discussion.*

Done. See deletion from line 269

*L258ff: The 'distal' samples not only seem to be different but they very obviously are.*

Sorry, there might be some British understatement in here. We have completely changed how we introduce these "distal samples" because it is now obvious to us that they aren't linked to the pingo like we thought they might have been. See lines 134 and then again in the results (lines 213 - 215) and discussion (lines 280 -286).

*L265ff: I find this paragraph confusing and the conclusions not convincing. It is indeed counterintuitive to expect that the influence of marine waters are higher the farther one comes from the sea. Furthermore, this conclusion is only supported if a part of the dataset (distal samples) is omitted from the analysis, but there is no justification given to do so.*

We felt we simply cannot ignore the strong influence of sea water upon the methane concentrations that is made obvious by our results. The likely causes of the counter-intuitive Cl- gradient are explained in our companion paper (Hornum et al, TC Discussions) and have been addressed in lines 295 - 301. We have also worked hard to justify the omission of the River Bed

Pingo Distal samples throughout the paper (see above), after realising that we had done a poor job of explaining it in the first manuscript. However, it is perfectly reasonable to do this in our opinion.

*Furthermore, it is unclear why the system is more diluted downstream by fresh groundwater from snow and ice melt.*
The system is increasingly diluted by fresh groundwaters because the valley is flanked by plateau highlands (line 301 and Figure 2a). See also response to next point.

*I understand, also from Fig. 4 that the sampled water originates from below the permafrost. In this case, the up-valley sites should be more influenced by melt water.*

Not really. The permafrost thicknesses are greater inland, so waters could be emerging from greater depths. This means the likelihood of denser, more saline springs increases inland. The issue is discussed at length in Hornum et al and so we feel uncomfortable extending our discussion to cover all of this hypothesis. However, we hope that the substantial amendments from line 295 cover the matter sufficiently without heavy reliance upon Hornum et al (In Review) at the request of the Editor.

*L286ff: The explanation of the variability in CH4 stable isotopes is unclear. Why should CH4 oxidation preferentially take place while the fluids are trapped below an ice lid and not during it's transport to the surface or after surface thaw?*

We argue that rapid switching in the source signature of the methane arriving at the pingo is unlikely because the system has a constant flow and long residence time. Rapid switching to 13C-enriched methane therefore seems most likely to be caused by the variable outburst cycle from beneath the ice blisters that form on the pingos. Methane gradually oxidises beneath the lid, then outbursts and refills once more, allowing "fresh" methane to mix with any residual "old" methane. Since the outbursts occur i) from different elevations on the pingo (thus emptying the ice blister to different degrees), and ii) after different storage times beneath it, the variability in 13C results from different mixing ratios between "fresh" and "old" methane at the time of sampling. It is unlikely that oxidation occurs during transport to the surface because the process is rapid compared to the storage time beneath the ice lid. Far less oxidation effects are therefore apparent in summer, when no such storage exists. See lines 311 – 323. Some oxidation prior to the ascent to the surface is also described in the discussion ending on line 340.

*To oxidize methane, an electron acceptor is needed, the respective microorganisms and liquid water but not stagnant water. And what might be the electron acceptor for methane oxidation? The fluids seem mostly oxygen free and low in sulfate.*

We appreciate the comments here. We will describe methanotrophy in a forthcoming paper, which has found that it occurs in the marine muds (at Lagoon Pingo and Forstehytte Pingo) but not the shale rock debris mantle on Innerhytte and River Pingos. We think it is beyond the scope of the present paper to describe this molecular work here, not least because we have found a novel organism at Lagoon Pingo. It will also support our assertions about why 13C-CH4 signatures at Lagoon+Forstehytte Pingos differ to Innerhytte+River Bed Pingos.

*L310: What means 'favorable thermodynamic conditions' in this context? Favorable for which process?*

The comment has been removed.

*L317ff: I understood from L286ff that the springs are frozen in winter. Please clarify.*

Partial freezing of the springs in winter forms an ice lid. This gradually expands upwards, then fractures and releases the spring water. It then flows over ice, releases methane and refreezes to form an icing. We regret not describing these processes further and have included appropriate text to describe what happens on lines 122 - 127

*L 331FF. Please give the reference for 'this paper'. Furthermore, clarify to which paper the newly introduced data from the 'neighboring lake' belong.*

Done (re proper reference of Hornum et al, TC Discussion paper). The additional data were the authors' own observations from just prior to submission and methane levels and dates of sampling are included in the text (line 360).

*L345: Pirk et al., 2017?*

Yes, corrected

*L347f: This calculation neglects aerobic methane oxidation, which might oxidize up to 100% of the methane before it is released into the atmosphere. Hence, the flux assumption from the springs is the upper limit of methane fluxes from the springs.*

We agree it is an upper estimate, but would like to point out that 100% removal is impossible in the system under study. We have changed the emphasis of the discussion to avoid direct discussion of emission (see earlier comments and lines 344 - 348)

*L363: Pirk et al. (2017 not 2018) report 'typically...a...seasonal budget of around 2gC m-2' (not 1 g C m-2) for the summer thaw season (1st June to 30th September) in Adventdalen. According to my calculation the annual flux from 4.7 km2 would then be about 12,600 kg methane yr-1 (not 6,040 kg methane yr-1).*

Citation corrected. The quote from the Pirk et al paper unfortunately refers to the median of two sites: Adventdalen and Zackenberg (in Greenland). The emissions from Zackenberg are greater than those from Adventdalen. Quick scrutiny of Figure 5 in Pirk et al (2017) clearly shows that all of the median values at Adventdalen lie below 2. It is therefore hard to justify using 2 gC m-2 y-1 as a spatially representative value (not least because it includes data from elsewhere). For this reason, I digitised Figure 5 in Pirk et al and determined the minimum and maximum median values from their three year study. These values were used to produce the range of likely emissions from wetlands for comparison with our emission estimates from springs. This range is presented in Table3.

*Furthermore, the winter fluxes are not considered in these estimates, which might be as high as the summer fluxes (see Zona et al., 2016).*

We have responded to this above.

*L375f: The meaning of this sentence ('The sensitivity...') is unclear.*

Appropriate amendments to this sentence and the next.

*L376ff: I cannot follow this calculation. Where does the number 50 L sec-1 come from? What is the Adventdalen terrestrial methane flux? In addition, why compare the total annual runoff of Adventdalen with the groundwater flux of 50 L- sec-1? The authors are aware that the methane concentration in surface melt water is several orders of magnitude lower than what they found in the springs with sub-permafrost fluids. This comparison is without meaning.*

We intended a straight-forward discussion of the sensitivity of potential methane emissions to a change in the water budget here. Every river has a baseflow largely driven by groundwater. Here we have just 1.6 L/s of sub-permafrost groundwater contributing to baseflow. Literature argues that the amount is likely to increase (citation of Victor Bense's work, who has been consulted directly). We therefore demonstrate that for an increase to 50 L/s then the flux of methane available for emission from the entire valley could increase by five times. Although 50 L/s seems high relative to the situation at the moment, it would still only represent 0.001% of total annual runoff. Most watersheds have a far higher degree of groundwater flow contributing to total runoff, but this is continuous permafrost terrain. Appropriate amendments have been made to the paragraph (lines 412 – 419) to help explain the purpose of this paragraph. Note that the criticism deserves less attention now that we have decided to not attempt a direct emission estimate.

*Table 2: Please also differentiate the "distal" samples from the River Bed Pingo*

Done

*Fig. 2 is difficult to read. Please give references for the published pore water data and please use units that make a comparison with the data in the tables possible (e.g. mgL-1 not µL mL-1)*

We appreciate that the diagram (Figure 4) needs time to understand properly and have tried once more to make it less cluttered with some edits. However, the units cannot be changed because the graph brings in pore gas analyses for comparison to our aqueous concentrations.

[revised manuscript text omitted]

---

## Editor Decision (ED1)

Dear Dr. Hodson,

Thank you for your initial responses to the open discussion.

The reviewers were generally favourable, but they want a number of items clarified. RC1's comments appear to be readily dealt with by improving the justification of the methods as you suggest. In contrast, RC2 provided a number of general and specific comments, but only the general comments were responded to though many of the specific comments are highly relevant. The reviewers provided many helpful suggestions that will help improve the paper, such as making justifications clearer and improving the background section. Ultimately, none of the critiques appear to be "off-base", and simply stem from some aspect that was not clear to the reader or some assumption or justification that appeared to the reader to be missing.

RC1 and RC2 do raise some questions about your exploration of emission fluxes. RC2 suggests that a more qualitative approach is warranted, and you seem amenable to this proposal, so please do. The data reported in the paper are on groundwater chemistry inferred from pingo springs and groundwater springs. There are no data on emissions, and both reviewers commented specifically on the estimation of methane emissions. Clearly the open-system pingos are a source of methane, but with the present level of uncertainty regarding actual emissions, I do support a change in the title that focuses on the data at hand and the study design. Perhaps something like "Springs from open-system pingos a source of highly concentrated sub-permafrost methane, Svalbard, Norway", or "Highly concentrated sub-permafrost methane sourced from open-system-pingo springs in Svalbard, Norway", or "Sub-permafrost groundwater from open-system-pingo springs a source of highly concentrated methane, Svalbard, Norway".

As a final comment, it is best to treat this manuscript as a stand-alone item, and not weigh heavily on Hornum et al. (2020) that is still in review. If there are background points that you can make in the present manuscript, please do.

In your Author's response to this decision I would like to read point-by-point responses to all of the reviewer's comments and questions and please indicate where any changes occur within the revised manuscript. Though the responses to the comments and questions may not alter the findings substantively, there enough potential changes to the manuscript that further review is warranted. Therefore, based on the reviewer's comments and your related discussions I will recommend publication subject to revisions (further review by editor and referees).

Thank you once again for contributing to *The Cryosphere Discussion*, and I sincerely look forward to reading a revised version of this manuscript.

Best regards,

Peter Morse

---

## Author Response (AR2)

**Response to Reviewers' and Editor's comments regarding resubmission of TC-2020-11**

Thanks for the clear guidance with respect to the minor revisions of the above re-submission. We removed the emission estimates and thus took the option to "greatly simplify" the Discussion and Conclusion sections. We are always improving our flux estimates and so we are working towards another manuscript to deal with these aspects of our work.

Here's what we did:

1) Changed the title to remove the word "emission"
2) Removed mention of the emissions from the abstract (last 2 sentences)
3) Changed the signposting at the end of the Introduction (last sentence)
4) Deleted the three paragraphs from Section 4.3 that dealt with emission quantification and made minor adjustments further down this section. We now use the remaining text to discuss the relevant processes that require consideration before the pingo emissions can be quantified. Note that in the last paragraph of the discussion we indicate that "the outflows …. Lack quantitative assessment" – thus indirectly indicating why no such emission estimates are given in this paper. We could be more direct, but see little point in explaining at length what the paper doesn't do.
5) Deleted the last 2 sentences from the Conclusion in order to remove emission emphasis
6) Deleted Table 3.
7) Corrected the reference list as a consequence of the changes above

With thanks,

Andy Hodson (on behalf of all co-authors)

---

## Editor Decision (ED2)

Dear Andrew Hodson et al.,

Re: tc-2020-11

Thank you again for your most recent revisions. The referees and I favour publication, though with some minor revision. Referee 1 still feels strongly that the data and findings regarding methane fluxes are overstated, and I tend to agree; the flux data are sparse but the Discussion is prominent. R1 provided further detailed comments on the flux estimates and would like to know how the discharge data presented in Table 3 are calculated and the uncertainties, how many measurements were taken, and, importantly, how representative are these estimates of annual flux given the very high variability of methane emissions from one of the pingos as demonstrated in an earlier publication (Hodson et al. 2019)? Given the degree of uncertainty, and if only a few measurements were made, the reader's confidence in the annual methane flux estimates is not high.

The main thing to keep in mind is that investigation of methane flux rates is not a stated purpose or objective in the introduction, nor are flux measurement methods elucidated in the study design, and nor are results presented. Please keep the major findings focused on the stated purpose, which guided the study design, field measurements, analytical work, and results: to investigate how "methane-rich fluids readily escape from beneath permafrost by exploiting the open system pingos that have formed following isostatic uplift and permafrost aggradation in Svalbard's fjord landscape."

It seems that the most obvious choice is to either greatly simplify related Discussion and Conclusions or provide substantially more data and information on methane flux in the manuscript, but you may have another solution. In the former case, please state that the annual methane flux estimates are simply a first-order approximation. Otherwise, annual flux estimation needs to be treated in more detail from start to finish.

In summary, this manuscript will be accepted for final publication subject to treating the lingering concern regarding annual methane fluxes.

I look forward to receiving your revised manuscript.

Best regards,

Peter

Hodson, A. J.; Nowak, A.; Redeker, K. R.; Holmlund, E. S.; Christiansen, H. H.; Turchyn, A. V., Seasonal dynamics of methane and carbon dioxide evasion from an open system pingo: Lagoon Pingo, Svalbard. Frontiers in Earth Science 2019, 7, (30)